# Granulocyte-colony stimulating factor controls neural and behavioral plasticity in response to cocaine

Erin S. Calipari[1,2], Arthur Godino[1,3], Emily G. Peck[1], Marine Salery[1], Nicholas L. Mervosh[4], Joseph A. Landry[1,4], Scott J. Russo [1], Yasmin L. Hurd[1,4], Eric J. Nestler[1,4] & Drew D. Kiraly [1,4,5]

Cocaine addiction is characterized by dysfunction in reward-related brain circuits, leading to maladaptive motivation to seek and take the drug. There are currently no clinically available pharmacotherapies to treat cocaine addiction. Through a broad screen of innate immune mediators, we identify granulocyte-colony stimulating factor (G-CSF) as a potent mediator of cocaine-induced adaptations. Here we report that G-CSF potentiates cocaine-induced increases in neural activity in the nucleus accumbens (NAc) and prefrontal cortex. In addition, G-CSF injections potentiate cocaine place preference and enhance motivation to self-administer cocaine, while not affecting responses to natural rewards. Infusion of G-CSF neutralizing antibody into NAc blocks the ability of G-CSF to modulate cocaine's behavioral effects, providing a direct link between central G-CSF action in NAc and cocaine reward. These results demonstrate that manipulating G-CSF is sufficient to alter the motivation for cocaine, but not natural rewards, providing a pharmacotherapeutic avenue to manipulate addictive behaviors without abuse potential.

[1] Fishberg Department of Neuroscience, Friedman Brain Institute, Icahn School of Medicine at Mount Sinai, New York, NY, USA. [2] Department of Pharmacology, Vanderbilt Center for Addiction Research, Vanderbilt University School of Medicine, Nashville, TN 37232, USA. [3] Department of Biology, École Normale Supérieure de Lyon, Lyon, France. [4] Department of Psychiatry, Friedman Brain Institute, Icahn School of Medicine at Mount Sinai, New York, NY, USA. [5] Seaver Autism Center for Research and Treatment, Icahn School of Medicine at Mount Sinai, New York, NY, USA. Erin S. Calipari and Arthur Godino contributed equally to this work. Correspondence and requests for materials should be addressed to D.D.K. (email: drew.kiraly@mssm.edu)

D rug addiction is a debilitating psychiatric condition characterized by dysregulated drug intake, enhanced motivation to both seek and take drugs, persistent use despite negative consequences, and recurring cycles of abstinence and relapse. Treatment of addiction to psychostimulants such as cocaine has proven particularly difficult despite extensive work characterizing the dopaminergic[1], glutamatergic[2], and neuronal signaling cascades[3] underlying the neurobiology of cocaine use. Even with our advances in knowledge, targeting these systems clinically in cocaine use disorder has proven to be difficult for a number of reasons, including problems with side effects, routes of delivery, or abuse potential of agents tested[4]. Thus, currently there are no FDA-approved pharmacotherapies for treatment of psychostimulant use disorders.

Studies of patients with cocaine use disorders have shown dysregulation of multiple peripheral cytokines—some of which correlate with extent of drug use[5], and addicts show altered immune system reactivity in response to drug cues[6]. While cocaine and other abused drugs are known to have effects on immune system functioning, only recently have studies begun to examine the mechanistic link between altered immune function and pathological substance use behaviors[7–10]. Here we aimed to define the complex interaction between cocaine use and cytokine signaling and how these factors alter reward, motivation, and economic decision making to drive cocaine addiction.

To this end, we characterize the regulation of innate immune system effector proteins in mice treated with cocaine. Via a broad multiplex screen of serum immune factors, we define several that are altered with cocaine exposure. However, while multiple immune factors are regulated by cocaine, only one—granulocyte-colony stimulating factor (G-CSF)—demonstrates upregulation in multiple treatment paradigms as well as correlation with an addictive phenotype. G-CSF has previously been shown to play a neuroprotective role in stroke[11,12], to delay degeneration in models of neurodegenerative disease[12,13], and to be important for learning and memory processes[14]. Here, we demonstrate that G-CSF is upregulated in the nucleus accumbens (NAc), a key brain reward region, by both cocaine and by the activation of medial prefrontal cortex (mPFC) to NAc projections, and is a potent regulator of both neuronal and behavioral response to cocaine. Together, these findings suggest that manipulation of G-CSF function may represent a new target for possible pharmacotherapies for patients with substance use disorders.

## Results

**Identification of immune targets altered by cocaine.** To identify potential soluble factors in blood associated with cocaine use, serum from mice treated with 10 daily doses of cocaine (Fig. 1a, b—20 mg kg$^{-1}$ i.p.) or 10 days of self-administration (Fig. 1c, d—0.5 mg kg$^{-1}$ for each infusion) was processed 24 h after the final dose for multiplex analysis of 32 cytokines, chemokines and growth factors. The values for each analyte, represented as the fold-change from the respective saline group, are shown as a heatmap in Fig. 1e (Experimenter admin: $n = 11$ saline, 15 cocaine; Self-admin: $n = 6$ saline, 10 cocaine; *$p < 0.05$; **$p < 0.01$). Raw pg/ml values for each analyte and exact $p$ values are available in Supplementary Data 1. To identify potential targets for further study, several factors were considered. First, if the effects were due directly to cocaine exposure, we expected the analyte to be significantly altered in the same direction in both experimenter and self-administered cocaine paradigms. Second, if it was related to behavioral response to cocaine, we expected the analyte to be correlated with cocaine sensitization and/or intake during the self-administration period.

Several analytes were significantly affected by cocaine exposure, but only two–G-CSF (Experimenter-Admin: two-tailed Student's $t$-test: $t_{(24)} = 2.48$, $p = 0.020$; Self-Admin: $t_{(9.4)} = 2.51$, $p = 0.032$) & interleukin-1α (IL-1α; Experimenter-Admin: $t_{(13.6)} = 2.19$, $p = 0.047$; Self-Admin: $t_{(13)} = 7.29$, $p < 0.0001$)— showed a statistically significant change in the same direction in both paradigms (Fig. 1e). Interestingly, while some analytes that were significantly regulated in only one paradigm showed a trend in the same direction in the other paradigm (e.g., KC, MIP-2), there were several that showed a relatively strong regulation in one cocaine administration paradigm, but no change at all in the other (for statistics and values, please see Supplementary Data 1). The biological significance of these discrepancies is not clear from these experiments, but our findings highlight important differences between the two administration paradigms.

As a marker of which differentially expressed analytes might be playing a role in behavioral responses to cocaine, we performed a correlation analysis between the serum level of each analyte with level of sensitization (in the i.p. injection paradigm), and the amount of average daily cocaine intake (in the self-administration paradigm). A heatmap of Pearson's r values for all analytes is presented in Fig. 1f (full correlation matrices with exact r and p values are available in Supplementary Data 2. Interestingly, only serum levels of G-CSF were increased by both experimenter-delivery (two-tailed Student's $t$-test $t_{(24)} = 2.48$, $p = 0.020$) and cocaine self-administration paradigms (Fig. 1g; $t_{(9.40)} = 2.51$, $p = 0.032$) and showed positive linear correlation with extent of both sensitization (Fig. 1h; Pearson's $r = 0.771$, $p = 0.025$) and self-administration (Fig. 1i; $r = 0.768$, $p = 0.026$). While G-CSF showed both regulation and correlation in both paradigms, this was not true for all regulated cytokines. Monokine-induced by gamma interferon (MIG, also known as CXCL9) was upregulated only after cocaine self-administration (Fig. 1j; $t_{(10.3)} = 3.74$, $p = 0.0036$), and showed positive correlation with levels of self-administration (Fig. 1l; $r = 0.766$, $p = 0.027$). However, levels of MIG were completely unaffected by experimenter-administered cocaine (Fig. 1j; $t_{(24)} = 0.177$, $p = 0.86$) and unrelated to cocaine-induced sensitization (Fig. 1k; $r = -0.361$, $p = 0.38$). Similarly, levels of IL-1α were significantly downregulated in both cocaine administration paradigms (Fig. 1m—exper-admin: $t_{(13.6)} = 2.19$, $p = 0.047$; self-admin.: $t_{(13)} = 7.29$, $p < 0.0001$), but showed no correlation with either behavior (Fig. 1n, o). Based on this evidence for the unique regulation of G-CSF by cocaine, it was chosen as the subject of more in depth functional analyses.

**Identifying the effects of G-CSF on neuronal activation.** Because we found that serum G-CSF levels were highly correlated with the extent of cocaine experience, we aimed to define where in the brain it might influence cocaine responses. Despite being a ~20 kDa protein, G-CSF is a soluble factor that readily crosses the blood brain barrier[15,16]. To identify G-CSF-responsive brain regions, we assessed transcript levels of the immediate early gene *c-Fos*, which is used as a marker of neuronal activation[17], after an acute i.p. injection of cocaine across several regions implicated in the motivation to self-administer cocaine: medial prefrontal cortex (mPFC), nucleus accumbens (NAc), dorsal striatum, ventral hippocampus, and basolateral amygdala (Fig. 2a). In the medial prefrontal cortex we saw the expected main effect of cocaine (Fig. 2b; two-way ANOVA $F_{(1,27)} = 16.41$, $p = 0.0004$), a main effect of G-CSF ($F_{(1,27)} = 5.243$, $p = 0.030$), and a trend towards a cocaine by G-CSF interaction ($F_{(1,27)} = 3.759$, $p = 0.063$). Post-hoc analysis revealed significantly enhanced *c-Fos* induction when comparing cocaine alone to cocaine + G-CSF (Holm–Sidak post-hoc: $p = 0.026$), but no effect when comparing saline to saline + G-CSF ($p = 0.80$). We saw a similar

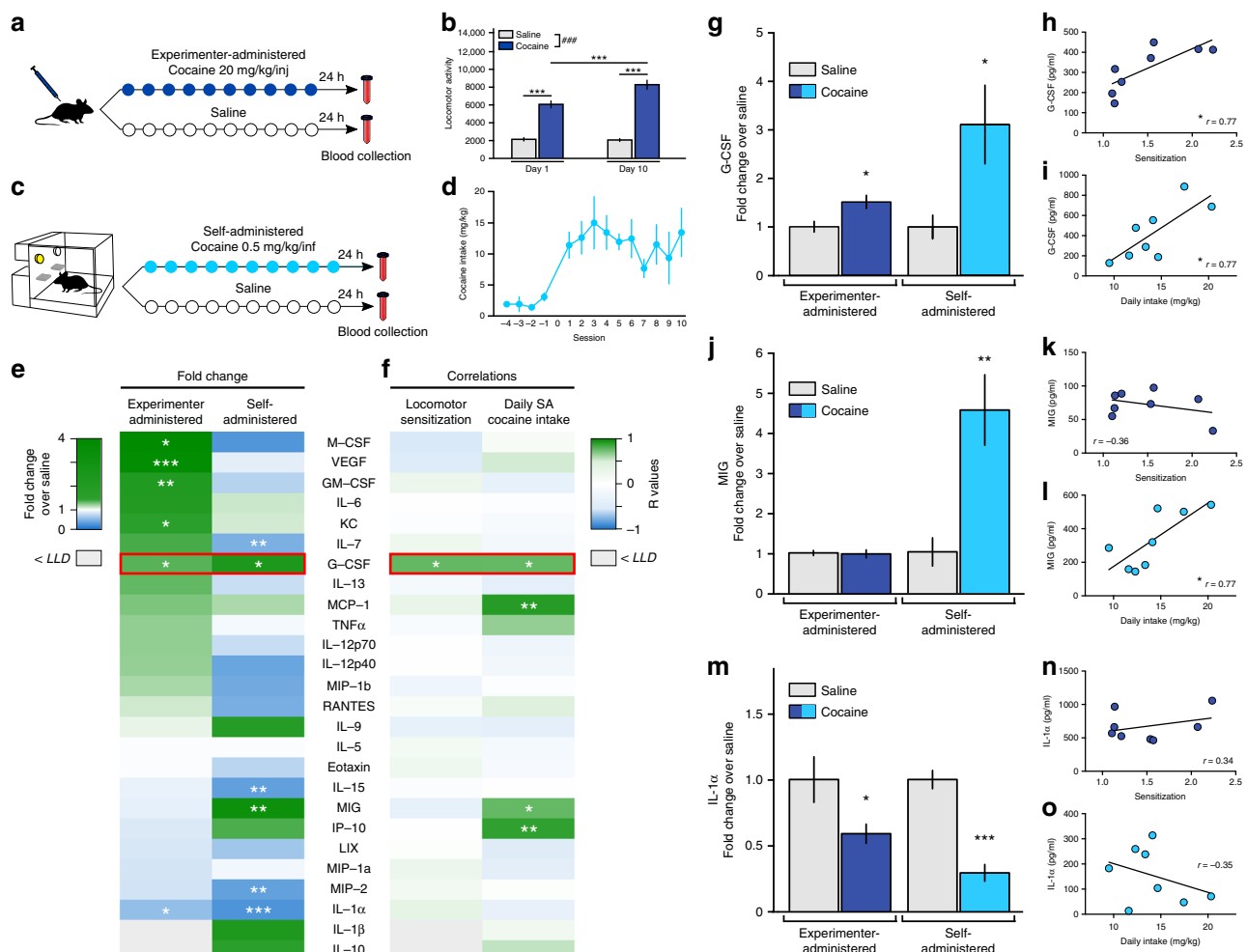

**Fig. 1** Serum multiplex analysis after self- and experimenter-administered cocaine in mice. **a** Timeline of experimenter-administered chronic cocaine injections. **b** Cocaine resulted in robust locomotor sensitization ($n = 10$ saline, 9 cocaine; two-way ANOVA–time: $F_{(1,17)} = 7.795$, $p = 0.013$; cocaine treatment: $F_{(1,17)} = 326.5$, $p < 0.0001$; interaction: $F_{(1,17)} = 9.035$, $p = 0.0080$; $p < 0.001$ vs control). **c** Timeline of cocaine self-administration in mice (saline: $n = 6$ saline; cocaine: $n = 10$). **d** Average daily intake of cocaine across self-administration sessions. **e** Multiplex serum analysis of 32 chemokines, cytokines, and growth factors after experimenter- or self-administered cocaine. For each analyte, the heatmap depicts fold-change values compared to the respective saline group. Raw pg/ml values for each analyte and exact $p$ values are available in Supplementary Data 1. **f** Correlation heatmap of individual analyte levels with either locomotor sensitization (Day 10/Day 1) or daily intake of cocaine. Exact r values for each analyte and exact $p$ values are available in Supplementary Data 2. **g** G-CSF is increased after both experimenter- (two-tailed Student's $t$-test; $t_{(24)} = 2.48$, $p = 0.020$) and self-administered cocaine ($t_{(9.40)} = 2.51$, $p = 0.032$), and G-CSF levels correlate with both **h** locomotor sensitization (Pearson's $r = 0.771$, $p = 0.025$) and **i** daily intake of self-administered cocaine ($r = 0.768$, $p = 0.026$). **j** MIG is increased only after self-administered cocaine ($t_{(10.3)} = 3.74$, $p = 0.0036$), and **k, l** individual MIG levels correlate only self-administered cocaine ($r = 0.766$, $p = 0.027$). **m** Levels of IL-1α are decreased after both experimenter-delivered ($t_{(13.6)} = 2.19$, $p = 0.047$) and cocaine self-administration ($t_{(13)} = 7.29$, $p < 0.0001$), however **n, o** IL-1α levels did not correlate with either behavior. Data represented as mean ± s.e.m. (*$p < 0.05$, **$p < 0.01$, ***$p < 0.001$ for Holm–Sidak post-hoc tests and $t$-tests)

pattern in the nucleus accumbens with a main effect of cocaine (Fig. 2c; $F_{(1,29)} = 33.62$, $p < 0.0001$), as well as a statistically significant main effect of G-CSF ($F_{(1,29)} = 6.803$, $p = 0.014$) which was driven by a G-CSF by cocaine interaction ($F_{(1,29)} = 5.215$, $p = 0.030$). As with the medial prefrontal cortex, in the nucleus accumbens of mice pre-treated with G-CSF there was significantly enhanced potentiation of *c-Fos* expression after cocaine compared to cocaine alone ($p = 0.0059$). This pattern was not seen in the other brain regions analyzed: there was a significant main effect of cocaine increasing *c-Fos* levels in the dorsal striatum (Fig. 2d; $F_{(1,28)} = 20.76$, $p < 0.0001$) and the ventral hippocampus (Fig. 2e; $F_{(1,24)} = 11.46$, $p = 0.0024$), but there were no main effects of G-CSF or any significant interactions, and in the basolateral amygdala we found no significant main effects or interactions (Fig. 2f).

There are extensive projections from the mPFC to the NAc—the only two regions that showed a significant interaction between G-CSF and cocaine—and further, the levels of *c-Fos* expression between these two regions was found to be correlated (Fig. 2g; Pearson's $r = 0.904$, $p < 0.0001$), suggesting the possibility that mPFC to NAc projections, already strongly implicated in behavioral effects of cocaine[18], may be playing a critical role in this process, possibly via glutamatergic projections driving further activation in the NAc.

**Regulation of G-CSF gene expression in the NAc and mPFC.** Given that G-CSF treatment controls patterns of neuronal activation in the NAc and mPFC in response to cocaine, we examined how treatment with cocaine might affect the expression of

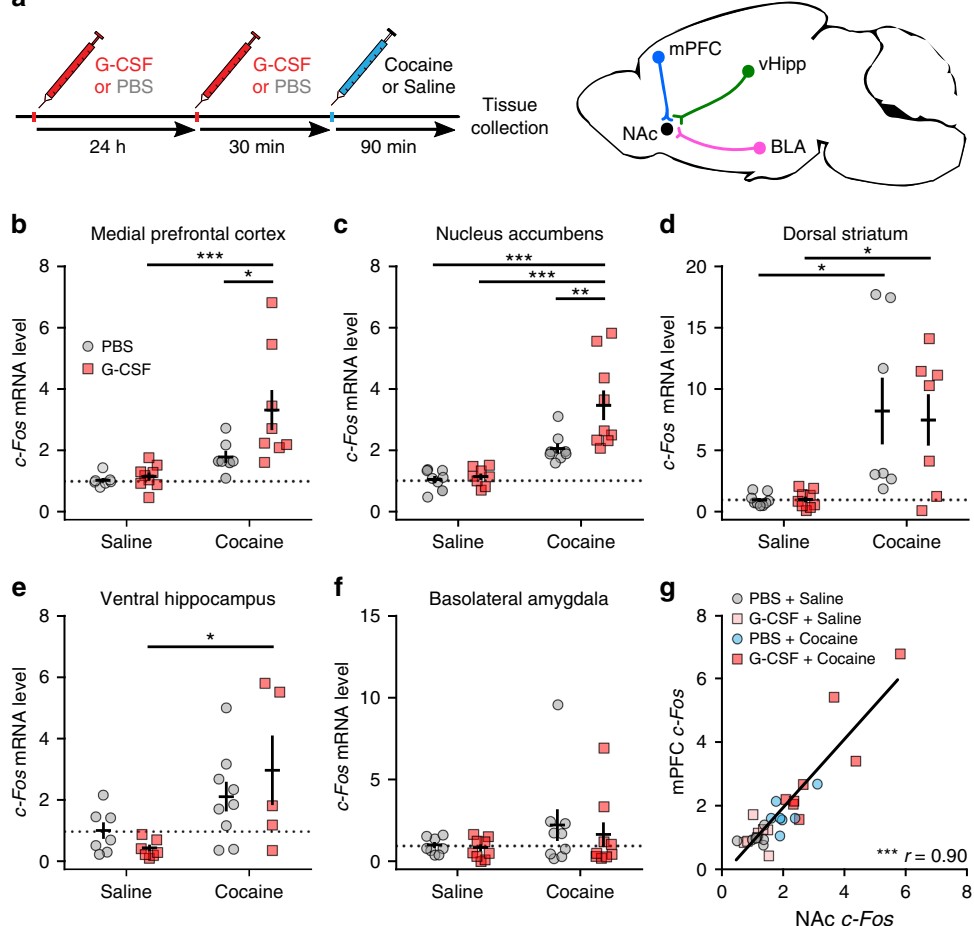

**Fig. 2** G-CSF potentiates cocaine-induced neuronal activation in specific brain regions. **a** Experimental Timeline (left). Mice were i.p. injected with G-CSF ($50 \, \mu g \, kg^{-1}$) or PBS 24 h and again 30 min before an injection of cocaine ($20 \, mg \, kg^{-1}$ i.p.) or saline and brain tissue was collected 90 min later. *c-Fos* expression was measured in critical brain regions involved in the motivation to self-administer cocaine (right): G-CSF enhanced cocaine-induced neuronal activation in the mPFC and NAc. **b** mPFC (two-way ANOVA – cocaine: $F_{(1,27)} = 16.41$, $p = 0.0004$; G-CSF: $F_{(1,27)} = 5.243$, $p = 0.030$; interaction: $F_{(1,27)} = 3.759$, $p = 0.063$), **c** NAc (cocaine: $F_{(1,29)} = 33.62$, $p < 0.0001$; G-CSF: $F_{(1,29)} = 6.803$, $p = 0.014$; interaction: $F_{(1,29)} = 5.215$, $p = 0.030$) **d** While cocaine increased neuronal activation in the dorsal striatum (two-way ANOVA –cocaine: $F_{(1,28)} = 20.76$, $p < 0.0001$), there was no added effect of G-CSF ($F_{(1,28)} = 0.05115$, $p = 0.82$) **e** Similar results were observed in the ventral hippocampus ($F_{(1,24)} = 11.46$, $p = 0.0024$; G-CSF: $F_{(1,24)} = 0.07447$, $p = 0.79$).
**f** *c-Fos* was not significantly induced by cocaine in the basolateral amygdala ($F_{(1,31)} = 2.463$, $p = 0.13$). **g** *c-Fos* expression levels were correlated between the NAc and mPFC (Pearson's $r = 0.904$, $p < 0.0001$). Data represented as mean ± s.e.m. (*$p < 0.05$, **$p < 0.01$, ***$p < 0.001$ for Holm-Sidak post-hoc tests)

G-CSF and its receptor locally within these regions. Transcript for G-CSF itself (*Csf3*) was significantly induced in both regions 90 min after an acute ($20 \, mg \, kg^{-1}$) i.p. injection of cocaine (Fig. 3a; two-tailed Student's *t*-test—NAc: $t_{(17)} = 2.60$, *$p = 0.019$; mPFC: $t_{(8)} = 3.06$, $p = 0.016$). In contrast, when animals were analyzed 24 h after a seven-day course of cocaine ($20 \, mg \, kg^{-1}$ per day, i.p.), levels of *Csf3* had increased even further in the NAc (Fig. 3b; NAc: $t_{(42)} = 3.57$, $p = 0.0009$), but were not significantly changed in the mPFC (Fig. 3b; $t_{(27)} = 1.15$, $p = 0.26$). Similarly, after this prolonged treatment with cocaine we found that levels of the G-CSF receptor were also increased in the NAc (Fig. 3c; $t_{(24.1)} = 2.71$, $p = 0.012$) but not in the mPFC (Fig. 3c; $t_{(20)} = 0.853$, $p = 0.40$). It is unclear as to why there were not changes in the mPFC, however, it is possible that changes in signaling from the receptor, not just relative expression levels could be playing a role. These findings, in conjunction with the *c-Fos* data, suggest strongly that the NAc is a crucial region for G-CSF signaling in response to cocaine.

Previous studies have shown that G-CSF and its receptor are expressed in forebrain neurons[16,19] and multiple subtypes of glial cells[20,21], however, the expression pattern in the NAc remains

unknown. The primary output neurons in the NAc are composed of two primarily non-overlapping populations of GABAergic medium spiny neurons (MSNs), defined by the predominant expression of D1 or D2 dopamine receptors. Thus, we performed immunohistochemistry to examine both expression levels of G-CSF and its receptor in these two subpopulations using mice that express tdTomato under the control of the D1 receptor promoter, and thereby identifying both D1$^+$ and D1$^-$ populations. G-CSFR was highly expressed in both D1$^+$ and D1$^-$ cell populations (Fig. 4a upper), suggesting expression in both cell types. In addition to expression in cells that morphologically resemble MSNs, expression was detected in surrounding glial cells.

Similarly, G-CSF itself seems to be expressed in a diverse array of cell types (Fig. 4a lower), and while there is clear peri-neuronal staining surrounding D1$^+$ MSNs, there was no specific cell pattern that emerged from the staining. Interestingly, when these same transgenic mice were treated with 7 days of experimenter-administered cocaine, there was no apparent shift in cell expression pattern of either G-CSF or its receptor (Fig. 4b). These results are consistent with findings from the periphery

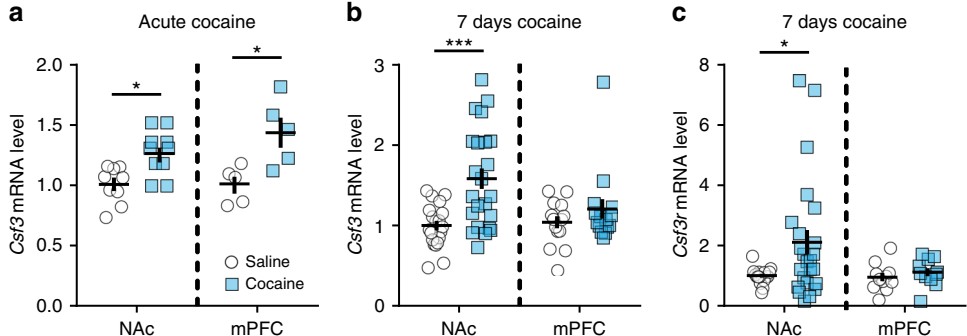

**Fig. 3** G-CSF-related gene expression is increased after cocaine. **a**, **b** mRNA levels of *Csf3* (G-CSF) in the NAc and mPFC after acute (**a**—two-tailed Student's *t*-test – NAc: $t_{(17)} = 2.60$, $p = 0.019$; mPFC: $t_{(8)} = 3.06$, $p = 0.016$) and 7 days of i.p. cocaine injections (**b**, NAc: $t_{(42)} = 3.57$, $p = 0.0009$; mPFC: $t_{(27)} = 1.15$, $p = 0.26$). **c** mRNA levels of *Csf3r* (G-CSF receptor) after 7 days of i.p. cocaine injections in the NAc ($t_{(24.1)} = 2.71$, $p = 0.012$) and mPFC ($t_{(20)} = 0.853$, $p = 0.40$). Data represented as mean ± s.e.m. (*$p < 0.05$, ***$p < 0.001$ for *t*-tests)

where G-CSF is expressed in myriad cell types including monocytes, endothelial cells, and fibroblasts[22–24], and suggest the possibility that the effect of G-CSF on behavior is occurring through effects on multiple cell types.

**G-CSF in the NAc is increased by mPFC to NAc stimulation**. While there were clear increases in G-CSF expression in NAc following cocaine exposure, we aimed to elucidate the underlying mechanisms. To this end, a two-part experiment was designed to determine if central or peripheral actions of cocaine were critical in G-CSF induction. First, we aimed to determine if increases in the activity of specific projection pathways in brain were capable of increasing central or peripheral G-CSF levels using designer receptors exclusively activated by designer drugs (DREADDs). Second, using a cocaine analog (cocaine methiodide) that does not cross the blood-brain barrier, we tested the effects of increased peripheral cocaine on central and peripheral G-CSF levels.

In the first series of experiments, a retrograde traveling CAV2 virus that drove the expression of Cre-recombinase was injected into the NAc. Cre-dependent expression of the excitatory $G_q$-coupled DREADD receptor was then induced in either the mPFC or the ventral tegmental area (VTA). This allows for pathway-specific stimulation of the mPFC to NAc or VTA to NAc projections (Fig. 5a). 7 days of daily clozapine-N-oxide (CNO) injections (1 mg kg$^{-1}$, i.p.) robustly increased the expression of *Csf3* in the NAc after mPFC to NAc, but not after VTA to NAc, stimulation (Fig. 5b One-way ANOVA - Main effect: $F_{(2,12)} = 13.4$, $p = 0.0009$; Sidak post-hoc: $p = 0.0037$). A similar effect was found for expression of the G-CSF receptor gene *Csf3r* (Fig. 5c; Main effect: $F_{(2,12)} = 8.14$, $p = 0.0058$; Sidak post-hoc: $p = 0.0093$). Stimulation of these two pathways did not have any significant effect on serum levels of G-CSF (Fig. 5d; Main effect $F_{(2,12)} = 1.82$, $p = 0.20$).

Next, we assessed whether the peripheral effects of cocaine, independent of any centrally-mediated actions, were sufficient to increase levels of G-CSF either in the serum or in the brain. To this end, mice were injected daily with cocaine methiodide–a charged analog of cocaine that does not cross the blood-brain barrier[25] (Fig. 5e). After 7 days of treatment, we found that cocaine methiodide did not affect transcript levels of *Csf3* (Fig. 5f; two-tailed Student's *t*-test—$t_{(12)} = 0.772$, $p = 0.45$) or *Csf3r* (Fig. 5g; $t_{(12)} = 1.11$, $p = 0.29$) in the NAc. Interestingly, serum levels of G-CSF in cocaine methiodide-treated mice were also unchanged (Fig. 5h; $t_{(12)} = 0.631$, $p = 0.54$). Taken together, these experiments suggest that G-CSF signaling in the NAc is regulated

by specific input pathways, and that peripheral upregulation of G-CSF by cocaine likely involves feedback signaling from the CNS.

**G-CSF enhances cocaine-induced locomotor sensitization**. To examine a possible causal link between systemic G-CSF and locomotor sensitization to cocaine, mice were injected i.p. with PBS or G-CSF (50 µg kg$^{-1}$) on the morning of each monitoring day one hour before testing (Fig. 6a). For the first 2 days, animals then received a saline injection and activity was monitored. Importantly, G-CSF on its own had no effect on locomotor behavior. Following repeated injections of cocaine, there were significant main effects of time (F $_{(6,42)} = 33.16$, $p < 0.0001$; $n = 4$ PBS, 5 G-CSF) and treatment with G-CSF (F$_{(1,7)} = 8.808$, $p = 0.021$), as well as a significant time×treatment interaction (F$_{(6,42)} = 3.942$; $p = 0.0032$). Holm–Sidak's post-hoc testing revealed that there were significantly different effects on days 2–5 of cocaine treatment (Fig. 6a), but not on the first day of cocaine —demonstrating that systemic G-CSF enhances locomotor responses to repeated injections of cocaine. It is noteworthy that in this set of experiments the PBS-treated animals seemed to show a delayed sensitization curve, however, the G-CSF treated animals were indeed increased when both groups seem to have reached a plateau at day 5. Importantly, given that G-CSF can mobilize immune cells from the bone marrow, we ensured that prolonged treatment with G-CSF at this dose does not cause infiltration of peripheral monocytes into the brain parenchyma (Supplementary Fig. 1).

**G-CSF alters preference formation specifically for cocaine**. While locomotor sensitization reflects behavioral plasticity associated with repeated cocaine exposure, it is dissociable from the subjective "rewarding" effects of cocaine. To assess whether systemic G-CSF also alters the ability of an animal to associate predictive contextual cues with the rewarding properties of cocaine, we used an unbiased conditioned place preference (CPP) assay in mice (typically used as a measure of cocaine "reward")[26]. We tested a range of cocaine doses beginning with a dose that does not lead to formation of preference in control animals (3.75 mg kg$^{-1}$) up to a relatively high dose (15 mg kg$^{-1}$) which does. For these experiments, animals were injected with G-CSF on each morning of the paradigm to maintain a high serum level of G-CSF. Treatment with G-CSF resulted in a significant leftward shift of the dose response curve (Fig. 6b; two-way ANOVA main effect of G-CSF: $F_{(1,36)} = 11.76$, $p = 0.0015$) leading to the formation of a robust preference for the lowest cocaine dose–which did not result in significant preference in control animals (Holm-Sidak

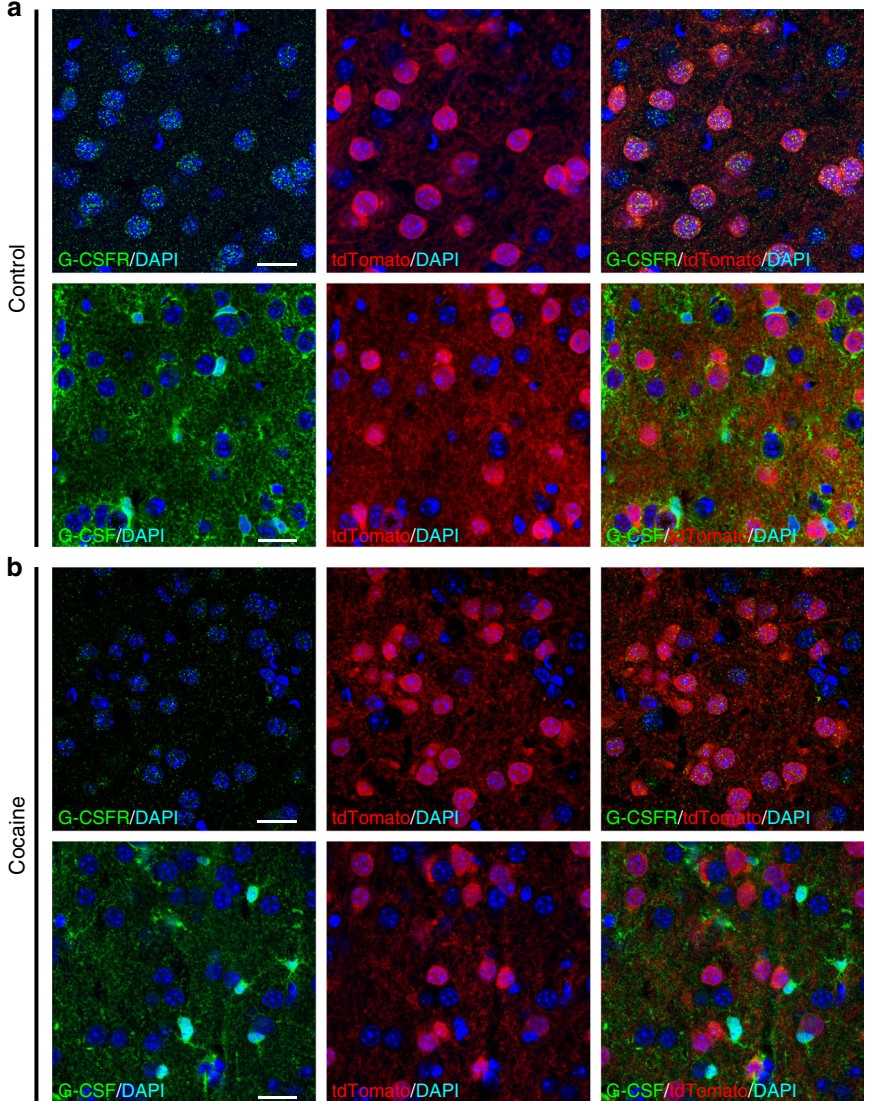

**Fig. 4** Detection of G-CSF and G-CSFR in the NAc of D1-tdTomato mice. Immunolabeling for tdTomato protein and GCSFR or GCSF in D1-tdTomato mice was performed to determine cell-type expression in the NAc. **a** Representative confocal images acquired in the shell of the NAc from control animals, demonstrating expression of G-CSFR (upper) and G-CSF (lower) in multiple cell types. **b** Representative images from mice treated with cocaine (20 mg kg$^{-1}$, i.p. × 7 days) again showing expression of G-CSFR (upper) and G-CSF (lower). For all images nuclei were counterstained with DAPI, tdTomato protein is labeled in red, G-CSF and G-CSFR are labeled in green. Scale bar = 20 µm

post-Hoc: $p = 0.028$; $n = 6$ PBS, 9 G-CSF) – and a significantly enhanced preference for the 7.5 mg kg$^{-1}$ dose (Holm–Sidak post-Hoc: $p = 0.035$; $n = 6$ PBS, 10 G-CSF), but no change at the highest dose (Holm–Sidak post-Hoc: $p = 0.38$; $n = 5$ saline, 6 G-CSF).

We next determined whether G-CSF has rewarding value on its own. In the first experiment, mice were injected with G-CSF each morning similar to the treatments used in Supplementary Fig. 2a, but in this case both chambers were paired with saline. Animals did not form any preference or aversion for either chamber (two-tailed Student's t-test: $t_{(7)} = 0.0862$, $p = 0.93$). In addition, we determined if animals would form associations between predictive contextual cues and G-CSF itself. We found no preference or aversion that resulted from G-CSF pairings (Supplementary Fig. 2b $t_{(8)} = 0.125$, $p = 0.90$).

We next determined if G-CSF's modulatory effects on cocaine reward learning were specific to cocaine or generalized to natural reward-related behaviors. We thus performed a two-bottle sucrose preference task in which animals choose between a water

bottle and another bottle with sucrose. Mice were treated daily with G-CSF. We found no difference between the two treatment groups (Supplementary Fig. 3a; two-tailed Student's t-test: $t_{(7)} = 0.348$, $p = 0.74$). Taken together, these results suggest that G-CSF enhances associative learning for drug rewards, but is not inherently rewarding or aversive, and does not affect preference for a natural reward.

**Causally linking G-CSF signaling to reward learning**. To determine whether G-CSF is also necessary for cocaine reward processing, we performed an experiment in which mice were injected i.p. with a G-CSF neutralizing antibody on each day of the CPP assay to reduce serum levels of G-CSF. Given that G-CSF crosses the blood brain barrier[15,16], but peripherally injected antibodies do not[27], this approach was intended to provide a readout of the effect of circulating G-CSF on the formation of place preference. There was a strong trend toward decreased preference in these animals, but there was no statistically

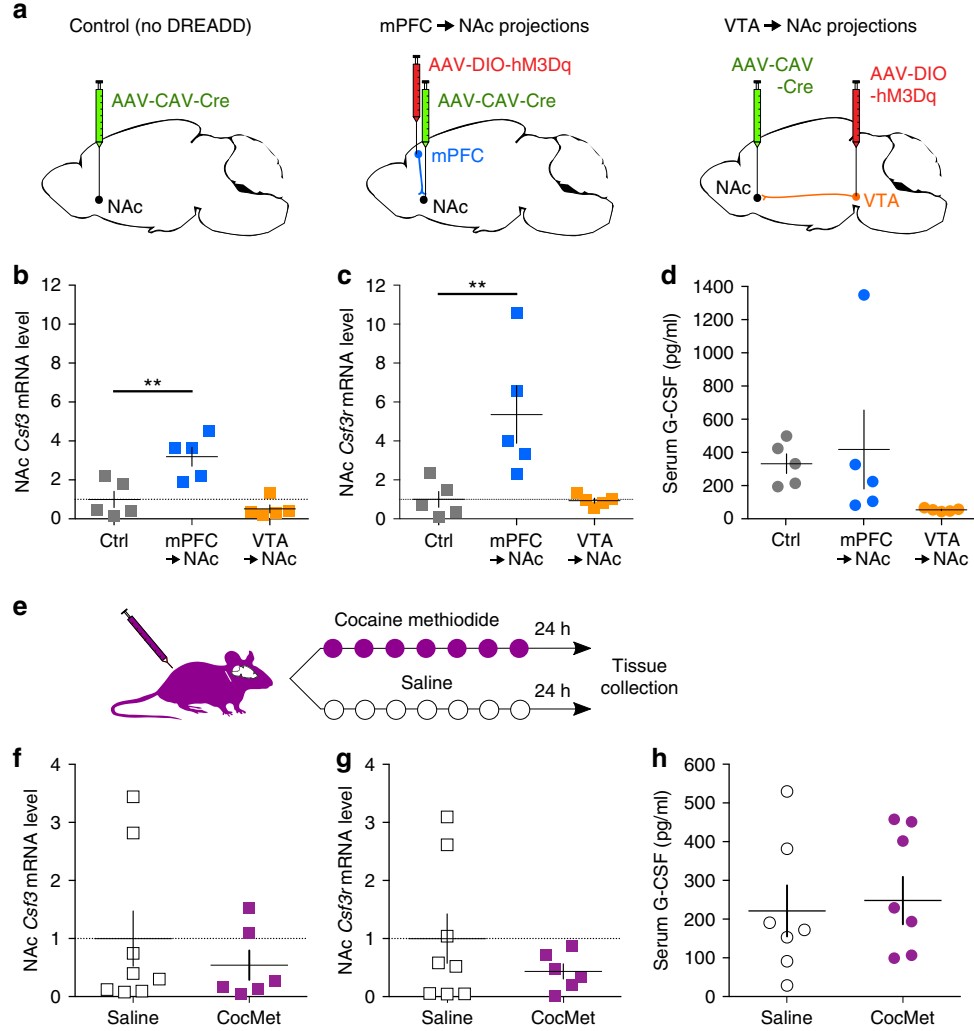

**Fig. 5** G-CSF levels are increased by the selective activation of mPFC to NAc projections. **a** Experimental design of projection-specific DREADD stimulation. Mice were injected with a retrograde traveling CAV2-Cre virus in the NAc and a Cre-dependent hM3Dq-DREADD virus in either the mPFC or the VTA to allow for the specific stimulation of either mPFC to NAc or VTA to NAc. **b** *Csf3* (G-CSF) mRNA levels in the NAc were increased after mPFC to NAc stimulation (one-way ANOVA; $F_{(2,12)} = 13.4$, $p = 0.0009$, Sidak post-hoc: $p = 0.0037$ vs control). **c** *Csf3r* (G-CSFR) mRNA levels in the NAc were increased only after mPFC to NAc stimulation ($F_{(2,12)} = 8.14$, $p = 0.0058$, $p = 0.0093$ vs control). **d** Peripheral G-CSF serum levels were not affected by stimulation ($F_{(2,12)} = 1.82$, $p = 0.20$). **e** Mice were injected i.p. for 7 days with cocaine methiodide (CocMet), a cocaine analog that does not cross the blood brain barrier, to assess the effects of peripheral cocaine on G-CSF. Cocaine methiodide chronic treatment had no effect on **f** *Csf3* (G-CSF) mRNA levels in the NAc (two-tailed Student's $t$-test; $t_{(12)} = 0.772$, $p = 0.45$), **g** *Csf3r* (G-CSFR) mRNA levels in the NAc ($t_{(12)} = 1.11$, $p = 0.29$), or **h** G-CSF serum levels ($t_{(12)} = 0.631$, $p = 0.54$). Data represented as mean ± s.e.m. (**$p < 0.01$ for Sidak post-hoc tests)

significant between-group difference (Fig. 7a; two-tailed Student's $t$-test: $t_{(13)} = 1.48$, $p = 0.16$). However, given that cocaine treatment bolsters G-CSF expression in the NAc (Fig. 3), we wanted to test the possibility that the signal needed to be neutralized more proximally to the site of action. For these experiments, we infused the same G-CSF neutralizing antibody or pre-immune IgG into the NAc using osmotic mini-pumps. When we tested place preference in these animals at the same dose, we saw that blockade of G-CSF signaling in this brain region with neutralizing antibody resulted in a significant reduction in the formation of cocaine place preference (Fig. 7b; $t_{(12.4)} = 3.75$, $p = 0.0026$).

**G-CSF alters the motivation to voluntarily consume cocaine.** CPP provides a measure of associative learning. Thus, changes in attention, learning mechanisms, or the rewarding properties of the stimulus can all result in increased place preference. To dissociate changes in reward, drug consumption, and motivation

from one another, we used drug self-administration in rats to understand how G-CSF alters the motivational properties of cocaine. Figure 8a outlines the timeline for these experiments. Rats were trained to acquire cocaine self-administration on a fixed-ratio one (FR1, 0.8 mg kg⁻¹ per infusion) schedule of reinforcement, and then split into two equally balanced groups with no significant differences in either acquisition (Fig. 8b) or cocaine intake (Fig. 8c) over the acquisition sessions before treatment with G-CSF or PBS, indicating that the groups were appropriately counterbalanced.

Rats were then treated with G-CSF (50 µg kg⁻¹ i.p.) on the evening following the last acquisition session and each morning 30 min before each self-administration testing session to ensure that G-CSF levels were adequately increased during testing. First, animals underwent testing using a behavioral economics threshold procedure, which is a within-session method used to assess an animal's motivation to self-administer a reinforcer in the face of increasing price (in this model price is equated to the number of

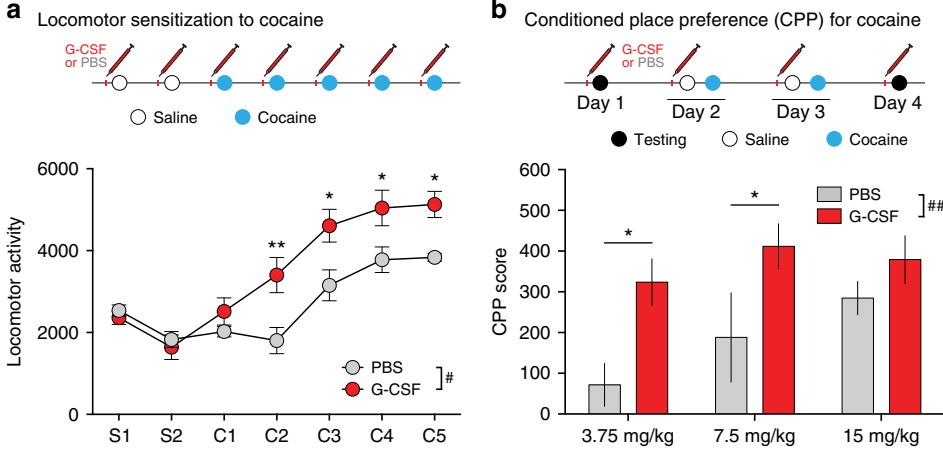

**Fig. 6** G-CSF enhances cocaine-induced locomotor sensitization and CPP. (**a**—top) Experimental timeline of locomotor sensitization to cocaine. Mice ($n = 4$ PBS, 5 G-CSF) were i.p. injected with G-CSF (50 µg kg$^{-1}$) or PBS 1 h before monitoring locomotor activity following an injection of saline or cocaine (7.5 mg kg$^{-1}$). (**a**—bottom) Locomotor sensitization to cocaine was increased in mice pre-treated with G-CSF (repeated-measures two-way ANOVA; time: F$_{(6,42)} = 33.16$, $p < 0.0001$; G-CSF: F$_{(1,7)} = 8.808$, $p = 0.021$; interaction: F$_{(6,42)} = 3.942$; $p = 0.0032$). **b** For cocaine conditioned place preference, mice were injected with G-CSF (50 µg kg$^{-1}$) or PBS 1 h every day before testing. Two-way ANOVA testing demonstrated a main effect of G-CSF (F$_{(1,36)} = 11.76$, $p = 0.0015$), and Holm-Sidak post-hoc testing demonstrated increased CPP in G-CSF-treated mice conditioned with 3.75 mg kg$^{-1}$ of cocaine (PBS $n = 6$; G-CSF: $n = 9$; $p < 0.05$ vs PBS), 7.5 mg kg$^{-1}$ (PBS: $n = 6$; G-CSF: $n = 10$; $p < 0.05$ vs PBS) of cocaine but not with 15 mg kg$^{-1}$ (PBS: $n = 5$; G-CSF: $n = 6$). (*$p < 0.05$, **$p < 0.01$ for Holm-Sidak post-hoc tests; #$p < 0.05$, ##$p < 0.01$ for two-way ANOVA main effects)

responses that the animal must emit to obtain 1 mg of drug)[28,29]. Figure 8d shows the dose-response curve indicating that G-CSF-treated animals lever-pressed more at lower doses as compared to PBS-treated animals. It is important to note that in this experiment price is inversely related to dose, where an increase in price (in responses per mg) is because the dose is low. Animals pre-treated with G-CSF increased the number of responses for lower doses of cocaine (Fig. 8d; repeated-measures two-way ANOVA: main effect of G-CSF: F$_{(1,15)} = 4.623$, $p = 0.048$).

The threshold procedure is particularly powerful because it allows for the application of economic principles to drug consumption[28]. In this task, animals self-administer cocaine on an FR1 schedule of reinforcement with no time outs. The dose starts high, therefore the price the animal has to pay in effort (relative price = number of responses that the animal must emit to obtain 1 mg of drug) is very low. As the session progresses, the dose is lowered, and animals' response rates will increase. While the responses increase, the relative drug consumption stays the same. This allows the data to be plotted as demand curves by plotting consumption as a function of price. Animals will maintain a preferred drug level when the cost is low and continue maintaining this level as the price of cocaine increases. Eventually the cost becomes too great and the animal will choose to reduce its intake. The inflection point of the curve occurs at P$_{max}$, the maximal price the animal is willing to pay for its preferred dose of drug. Animals pretreated with PBS had a P$_{max}$ of ~150 responses per milligram (Fig. 8e). Animals pretreated with G-CSF had significantly higher P$_{max}$ at around 225 responses per milligram (Fig. 8f, h; two-tailed Student's *t*-test: t$_{(14)} = 2.002$, $p = 0.033$). Group demand curves were significantly different (Fig. 8g; repeated-measures two-way ANOVA: main effect of price: F$_{(9,135)} = 72.88$, $p < 0.0001$; main effect of G-CSF: F$_{(1,15)} = 5.036$, $p = 0.04$), further substantiating the interpretation of an increase in motivation induced by G-CSF.

In addition to the G-CSF-induced increase in motivation, there was an increase in Q$_0$, a measure of cocaine intake (Fig. 8i; t$_{(14)} = 2.374$, $p = 0.016$). Further supporting that G-CSF increased consummatory behavior, intake was also increased on FR1 schedule of reinforcement (Fig. 8j; t$_{(15)} = 2.87$, *$p = 0.12$). To assess how G-CSF affected the appetitive and consummatory

responses controlled by natural rewards, we performed a similar threshold task using food reward, in which rats were treated with G-CSF or PBS prior to the threshold sessions. Interestingly, G-CSF did not affect Q$_0$ or P$_{max}$ (Supplementary Fig. 3b, c). Altogether, G-CSF increases both the motivation to self-administer cocaine and the amount of cocaine an animal wants to consume while not affecting either parameter for natural rewards.

## Discussion

Here, using quantitative serum multiplex analysis, we identify G-CSF as an innate immune factor upregulated peripherally and within the NAc by multiple paradigms of cocaine administration. By using in vivo manipulations of G-CSF function, we identify its role in promoting neural responses to cocaine, as measured by upregulation of *c-Fos*, and find that G-CSF is a potent enhancer of behavioral responses to cocaine, but not natural rewards, in multiple behavioral tasks. Our data suggest that the locus of effect for G-CSF in both neural and behavioral responses to cocaine is the NAc, a limbic structure well characterized for its role in addictive behaviors.

Of the many analytes we evaluated initially, G-CSF was chosen for further study as it was found to be upregulated by both experimenter-administered and self-administered cocaine, and in a manner that correlated with behavioral output in response to the drug. Interestingly, a number of other cytokines were upregulated in one paradigm or another—an effect that may be due to myriad causes. First, the pattern of intake in experimenter-administered cocaine is that of a large bolus of drug given over a few seconds. Self-administering animals took less total drug over the course of the experiment, and the infusions were in small doses spaced over several hours. The longer-spaced intake might explain why some analytes such as IL-6 and KC showed matching directional changes in both paradigms, but with smaller amplitude of effect in self-administering animals. The bolus dosing that is achieved with experimenter-administered cocaine results in a larger surge of sympathetic nervous system activity, which has been shown to alter immune responses[30]. In addition, the voluntary nature of self-administering cocaine is complicated and

results from combinatorial effects of multiple circuits involved in decision making, motivation, and assessing subjective value of the drug. In experimenter-delivered paradigms all animals are given the same amount of drug without volitional control, which can alter the subjective perception of the drug. This is important, as pattern, drug level, and preferred intake level have been shown to play a critical role in drug-induced plasticity and neural adaptations[31,32].

While this is one of the first studies to directly and causally link a soluble immune factor to motivation for drugs of abuse, this is not the first study to demonstrate neuroimmune interactions in response to cocaine. Cocaine has long been known to have effects on immune function[33], and is thought to increase the risk of HIV infection even when controlling for behavioral covariates in drug-abusing populations[34,35]. Studies in humans have shown alterations in cytokines long after cessation of drug use, with some of the abnormalities correlating with extent of previous use[5,36]. In addition, treatment-seeking cocaine users show enhanced serum expression of pro-inflammatory tumor necrosis factor alpha (TNF-α) and decreased anti-inflammatory IL-10 in response to either stress or drug cues[6].

These human studies provide correlational evidence of clinical relevance of neuroimmune interactions in the behavioral response to cocaine, but mechanistic studies examining the basis of such interactions have only begun to be published[7,10]. A recent study by Northcutt and colleagues used in silico modeling and binding assays to demonstrate cocaine activation of the toll-like receptor 4 (TLR4)[8]. Activation of TLR4 and downstream production of interleukin 1 beta (IL-1β) in the VTA was found to be crucial for striatal dopamine release, expression of cocaine place preference and self-administration[8]. In our peripheral cytokine assays, we found IL-1β was expressed at very low levels after experimenter-administered cocaine, and showed a trend toward an increase after self-administered cocaine, but no correlation was seen with behavioral output. Another recent study found that TNF-α specifically released from microglia in the NAc was a key modulator of neuronal and behavioral plasticity in response to cocaine, and served as a brake on the development of locomotor sensitization[9]. We did not find significant upregulation of TNF-α in our samples, but there was a strong trend toward positive correlation between TNF-α levels and cocaine self-administration. It is important to note that our assays were serum based, and the studies from Lewitus et al.[9] and Northcutt et al.[8] were both focused on central cytokine signaling, with the Lewitus study focusing specifically on microglial-derived TNF-α. In a number of other studies, G-CSF has been shown to reduce CNS levels of both TNF-α and IL-1β[37–39], and one possible mechanism by which it may be having its effect in our study is via affecting the production of other immune mediators. Future studies will focus on how G-CSF affects other innate and adaptive immune mediators in ways that influence behavioral responses to drugs of abuse.

Once G-CSF was identified as a cytokine altered by cocaine exposure, the next goal was to determine the role of G-CSF in modulating cocaine-associated behaviors and to define the precise neural mechanisms that control these effects. The receptor for G-CSF has been shown to be expressed on neurons throughout the CNS[12,16,19], suggesting a role as a modulator of neuronal activity. Indeed, G-CSF has been shown to induce c-Fos, a marker of cellular activation, in midbrain neurons[40]. In our study, mRNA levels of G-CSF and its receptor were induced in the NAc after acute and chronic cocaine treatment, suggesting a role for mesolimbic system modulation in these cocaine-mediated behavioral effects. This prompted further investigation into the cellular localization of G-CSF and its receptor in the NAc. We observed expression in D1 and D2 MSNs, as well as in glial cells,

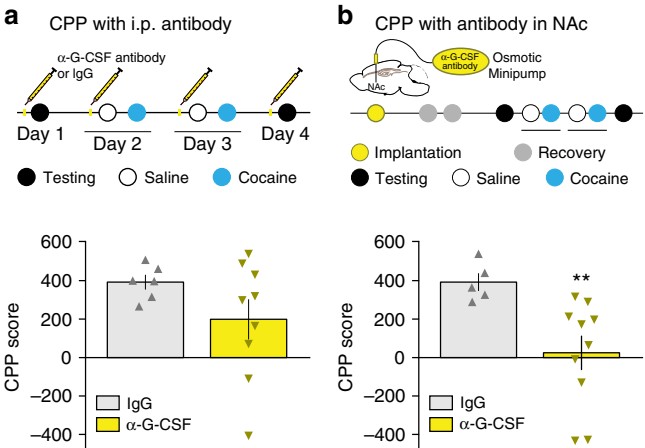

**Fig. 7** Neutralization of central G-CSF signaling reduces conditioned place preference. **a** To determine the role of circulating G-CSF in behavior, mice (IgG: $n = 6$; α-G-CSF antibody: $n = 9$) were i.p. injected with anti-G-CSF antibody (10 μg) or pre-immune IgG control antibody 1 h every day before testing for CPP at 7.5 mg kg⁻¹ of cocaine. Systemic anti-G-CSF antibody did not significantly affect cocaine CPP (Two-tailed Student's t-test: $t_{(13)} = 1.48$, $p = 0.16$). **b** To test the effects of blocking signaling in the NAc anti-G-CSF antibody or pre-immune IgG (1 μg/side) was infused into the NAc via continuous osmotic minipump before and during testing for CPP at 7.5 mg kg⁻¹ of cocaine (IgG: $n = 5$; α-G-CSF antibody: $n = 10$). NAc infusion of anti-G-CSF antibody blocked cocaine CPP ($t_{(12.4)} = 3.75$, $p = 0.0026$). Data represented as mean ± s.e.m (**$p < 0.01$ for t-test)

showing that receptor localization alone likely did not confer any pathway-specific actions.

One important feature of G-CSF, defined within this study, is its ability to modulate cocaine-induced behavioral outputs independent of any rewarding effects of G-CSF itself. Further, G-CSF did not alter sucrose preference, food-maintained responding on low effort schedules, or motivation for food rewards, suggesting that G-CSF could provide a powerful tool to modulate addictive behaviors with limited side effects on naturally rewarding stimuli. The disparity between the modulation of drug and food is particularly interesting; however, not surprising. While motivation for food and drugs of abuse are modulated by some of the same circuitry[41], motivation for food has many redundant pathways that can modulate motivation[42,43]. Further, homeostatic pathways driving food intake are tightly regulated and are modulated by internal state and need[44,45]. It is possible that G-CSF is only capable of modulating reward behaviors that are particularly salient. It would be interesting in the future to conduct experiments that modulate satiety state to determine if G-CSF is capable of altering motivation for food when need is high.

Our studies provide insight into the central role of G-CSF in controlling the neural and behavioral responses to cocaine and define the locus of these effects to specific projection pathways. G-CSF signaling in the NAc was critical in modulating the behavioral responses to cocaine, as a G-CSF-neutralizing antibody injected peripherally resulted in only a slight decrease in the formation of cocaine conditioned place preference, but infusion directly into the NAc robustly decreased behavior. Further, c-Fos has been shown to be robustly induced in NAc by acute cocaine exposure[3,17,46], and we found that pre-treatment with systemic G-CSF significantly potentiated this induction, suggesting that G-CSF acts to increase the activity of cells in the NAc either directly or through modulation of inputs into the region. Interestingly, c-Fos was also increased in the mPFC and the increased activity was

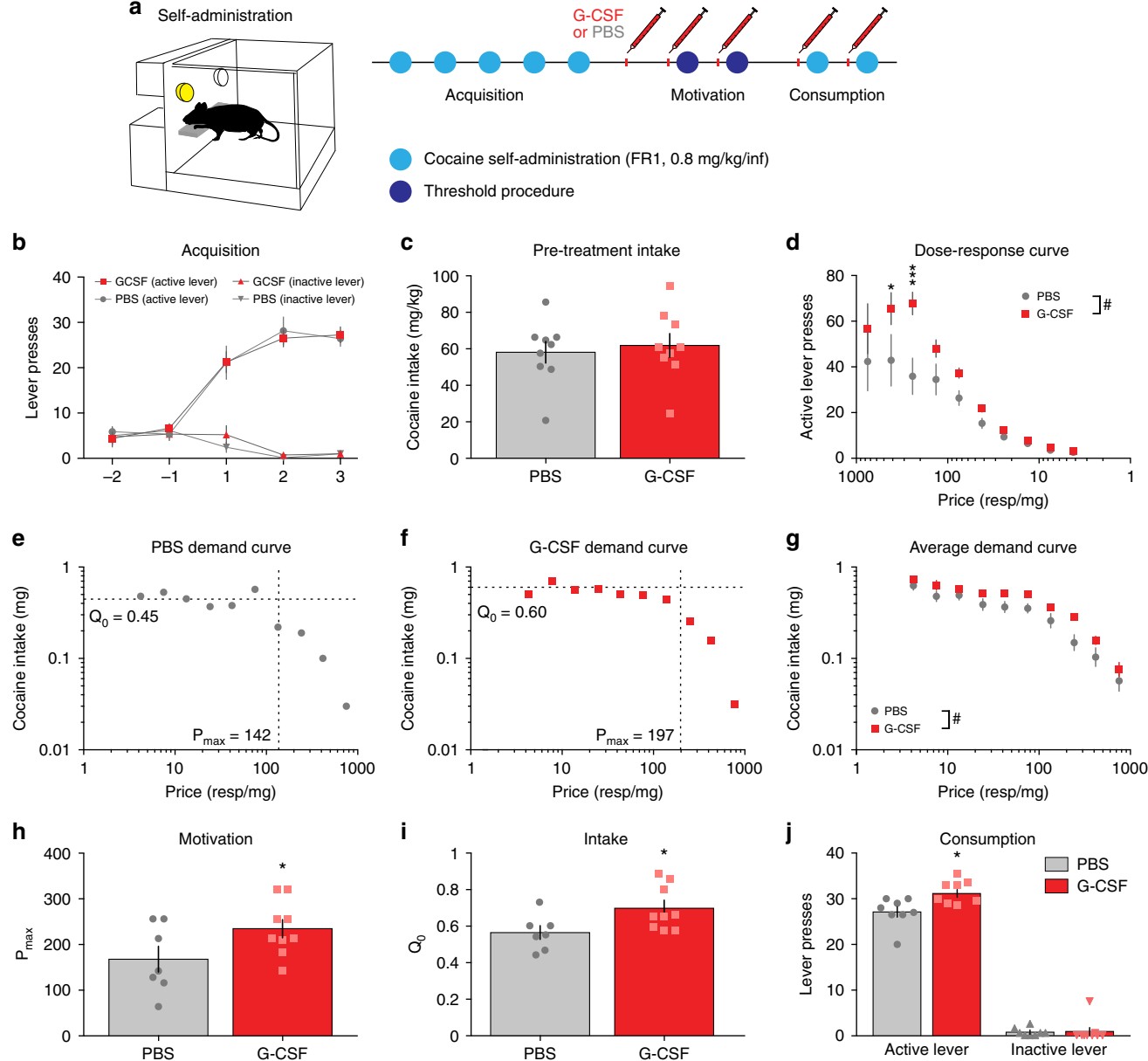

**Fig. 8** G-CSF increases the motivation to self-administer cocaine. **a** Experimental timeline. Rats were trained to self-administer cocaine on a fixed-ratio one (FR1) schedule of reinforcement. Animals went through two behavioral tasks: the threshold procedure to assess motivation, and FR1 self-administration to assess consumption. **b** Acquisition for the two experimental groups before treatment showing that there were no significant differences before the experiment (two-way ANOVA: time: $F_{(4,64)} = 69.11$, $p < 0.0001$; group: $F_{(1,16)} = 0.00147$, $p = 0.97$; interaction: $F_{(4,64)} = 0.1985$, $p = 0.94$). **c** Cocaine intake did not differ in the two groups before G-CSF or PBS treatment (Student's $t$-test; $t_{(16)} = 0.427$, $p = 0.34$). **d** Dose-response curves. G-CSF pretreatment increased responding for lower doses of cocaine indicating an upward shift in the dose–response function (two-way ANOVA; price: $F_{(9,135)} = 42.06$, $p < 0.0001$; G-CSF: $F_{(1,15)} = 4.623$, $p < 0.05$; interaction: $F_{(9,135)} = 2.616$, $p < 0.01$). **e, f** Representative demand curves, plotting consumption of cocaine as a function of price, from a PBS-treated control and G-CSF-treated animal. **g** Averaged demand curves from both groups (two-way ANOVA; price: $F_{(9,135)} = 72.88$, $p < 0.0001$; G-CSF: $F_{(1,15)} = 5.036$, $p = 0.40$; interaction: $F_{(9,135)} = 0.9677$, $p = 0.47$) **h** $P_{max}$ is increased in animals treated with G-CSF (Student's $t$-test; $t_{(14)} = 2.002$, $^*p = 0.033$). **i** $Q_0$ levels are increased in G-CSF treated animals ($t_{(14)} = 2.374$, $^*p = 0.016$). **j** Intake was also measured using a fixed ratio one schedule of reinforcement. Animals treated with G-CSF took more cocaine injections in a three-hour session than their PBS-treated counterparts (Student's $t$-test for active lever presses; $t = 2.866$, df = 15, $p = 0.012$). Data represented as mean ± s.e.m ($^*p < 0.05$, $^{**}p < 0.01$, $p < 0.001$ for Holm–Sidak post-hoc tests and $t$-tests; $^{\#}p < 0.05$ for two-way ANOVA main effects)

correlated between the mPFC and NAc regions, suggesting that activity of this specific projection pathway may play a role.

We conducted two separate experiments to determine how cocaine was acting to increase G-CSF levels within the brain. First, we used a brain impenetrable cocaine analog, cocaine methiodide, to see if central actions of cocaine were necessary to increase G-CSF levels. We did not see increases in either central

or peripheral levels of G-CSF, suggesting that the increase in serum G-CSF is dependent on some component of signaling from the CNS and that cocaine induction of G-CSF in NAc requires central actions of the drug.

Second, we used pathway-specific DREADD stimulation to determine if activation of mPFC to NAc or VTA to NAc circuits in isolation were sufficient on their own to change G-CSF levels.

We found that G-CSF and its receptor in the NAc were induced only by stimulation of the glutamatergic mPFC to NAc pathway, but not the VTA to NAc pathway. Interestingly, G-CSF is also robustly and rapidly upregulated in the CNS in response to stroke or other neuronal insults in which there is rapid upregulation of glutamatergic signaling[16], and has repeatedly been shown to be neuroprotective in models of stroke and neurodegenerative conditions[11,12,20]. In the case of cocaine abuse, this sets forth an interesting positive feed-forward model where cocaine-induced activation of the mPFC to NAc pathway is capable of increasing brain G-CSF levels. G-CSF in the brain can then enhance cocaine-induced neuronal activation in these regions, further augmenting the rewarding and reinforcing properties of cocaine itself.

While we have defined a causal role of G-CSF in several models of cocaine reward, motivation, and the plasticity induced by repeated exposure, one of the most interesting things about G-CSF was its lack of effect on its own. G-CSF did not induce CPP or lead to a large increase in c-Fos expression in brain, but rather played a role in enhancing stimulus-driven activation and learning mechanisms. In our behavioral assays, G-CSF augments the formation of locomotor sensitization in response to cocaine, leads to enhanced formation of conditioned place preference, and enhances motivation to self-administer cocaine, but does not affect preference for, or intake of, natural rewards. Other studies have demonstrated significant behavioral and cognitive effects of G-CSF in animal models. Mice with a constitutive knockout of G-CSF have significantly impaired performance on a Morris Water Maze task, and do not form hippocampal LTP in ex vivo slice preparations[14]. Treatment of animals with exogenous G-CSF during a spatial learning task enhances acquisition of the task[47], and cognitive impairments in animal models of Alzheimer's pathology are slowed or reversed by treatment with G-CSF[37,48]. It is possible that signaling downstream of G-CSF enhances synaptic connectivity in ways that promote associations of context or cues with a rewarding stimulus such as cocaine.

Giving further credence to this idea is our self-administration data showing that G-CSF modulates performance, without disrupting it, on tasks where animals are making complex decisions about economic value. This is important for two reasons. First, it is important that simple measures of reward are not contaminated with reductions in arousal, attention, or cognitive performance. For tasks like conditioned place preference, this can be the case as it depends on associative learning mechanisms. Thus, animals that cannot attend to cues or have impaired memory will exhibit behaviors that look like reduced reward. However, with behavioral economics tasks, animals have to make complex decisions in a changing environment; thus, a drug that alters factors other than motivation and reward value would prevent animals from effectively completing the task. G-CSF did not alter the ability of animals to complete the self-administration tasks, suggesting that it modulates reward and reinforcement directly. Second, a major shortcoming of some behavioral paradigms is that findings have not translated to the clinic. Behavioral economics provides a potential tool to improve this translation. Indeed, $P_{max}$ is disrupted in drug-addicted human subjects[49], and is directly correlated with addiction severity, where increased $P_{max}$, and reduced sensitivity to cost, are characteristics of drug addiction. Thus, treatments that allow individuals to reduce the value that they place on cocaine, without disrupting decision-making processes, are critical to develop cocaine treatments. G-CSF has these components, and here we show that G-CSF can directly modulate $P_{max}$ and that reducing G-CSF levels reduces cocaine reward, providing a potential therapeutic target for the treatment of drug addiction.

Altogether, this work has defined the role of a soluble immune factor in complex behavior related to addiction, defined the mechanistic and causal role G-CSF in this process, and provides a potential pharmacotherapeutic target to improve treatment outcomes in addicted individuals. While these studies suggest that increases in G-CSF may be maladaptive in the setting of drug abuse, targeted manipulations of G-CSF and its downstream signaling pathways may represent tractable treatment strategies. These findings represent an important advance in the understanding of the pathophysiology of, and possible treatment for, substance use disorders. Identification of an innate immune effector that is capable of affecting behavioral responses to cocaine has high relevance for potential clinical translation, and recombinant G-CSF has been FDA-approved for many years to treat neutropenia[50] and is now in clinical trials for both stroke[51] and Alzheimer's disease[52]. The idea of using an immune modulator to treat psychiatric pathology is not novel, and early stage clinical trials of pharmacotherapies aimed at targeting the immune system have shown promise in treating major depressive disorder[53]. Treatment with a G-CSF modulator would have the distinct advantage that it may be harnessed to reduce drug taking while ostensibly having no abuse potential on its own—a known confound in many previous trials for psychostimulant use disorders[4]. While further study and care will be necessary before transitioning these findings toward a clinical population, we posit that this study represents the possibility of a highly tractable strategy for clinical translation with a good potential for success.

## Methods

**Animals**. Male C57BL/6 J mice (7wks old ~20–25 g; Jackson Laboratories, Bar Harbor, ME) and Sprague-Dawley rats (350–375 g; Envigo, Somerset, NJ) were housed in the animal facilities at Mount Sinai. For all experiments other than self-administration, mice were maintained on a 12:12 h light/dark cycle (0700 hours lights on; 1900 hours lights off). Animals were provided with food and water ad lib throughout the experiments. For self-administration studies, mice and rats were maintained on a 12:12 h reverse light/dark cycle (0700 hours lights off; 1900 hours lights on). Animals were food restricted to 95% of free feeding weight throughout the duration of self-administration experiments. Food restriction and reverse light cycle serve to improve consistency of the self-administration task. All animals were maintained according to the National Institutes of Health guidelines in Association for Assessment and Accreditation of Laboratory Animal Care accredited facilities. All experimental protocols were approved by the Institutional Animal Care and Use Committee at Mount Sinai.

**Serum multiplex analysis**. Adult male mice were treated with experimenter-administered (20 mg kg⁻¹ per day i.p.) or self-administered (0.5 mg kg⁻¹ per infusion i.v. × 2 h/day) cocaine for 10 days during which locomotor sensitization behavior and self-administration intake were monitored respectively. Mice were analyzed 24 h after the final dose of cocaine and trunk blood was collected. Blood was allowed to sit at room temperature for at least one hour to allow for full clotting. Samples were then spun at 1200×g for 15 min and the supernatant was saved as serum. Multiplex analysis was performed using the commercially available mouse cytokine/chemokine magnetic bead panel (Millipore MCYTMAG-70K-PX32) on a Luminex 200 multiplex immunoassay system at the Human Immune Monitoring Core at Mount Sinai. Assays were quality control checked for fit to a standard curve, and for coefficient of variation of < 2.5%. For analytes below the lower level of detection (LLD) of the assay, values were imputed as 0.5* LLD as has been reported previously[54]. There were six analytes for which > 60% of values were below the lower limit of detection (IFN-1, IL-2, IL-3, IL-4, IL-17, LIF) and these were excluded from further analysis. For correlational analysis serum levels of each analyte were compared to the level of locomotor sensitization exhibited (Day10/Day1 Locomotor activity) or to the average daily self-administration during stable self-administration (Days 5–10).

**qPCR**. For biochemical experiments mouse brain punches were taken from animals sacrificed either 90 min after an acute injection of cocaine (20 mg kg⁻¹) or 24 h after 7 daily injections of cocaine (20 mg kg⁻¹). Brain punches were analyzed following the described treatments and brain tissue was rapidly dissected on ice and frozen on dry ice. RNA isolation and qPCR were performed as described previously[55,56]. Briefly, RNA was extracted by homogenizing tissue in Qiazol reagent (Qiagen) and purified using RNeasy micro kits from Qiagen according to manufacturer protocols. RNA concentration and quality were assessed using a Nano-Drop spectrophotometer (Thermo). Reverse transcription was performed using iScript (BioRad) according to manufacturer protocols. qPCR using SYBER green master mix (Quanta) was carried out using an Applied Biosystems 7900HT cycler with the following parameters: 2 min at 95 °C; 40 cycles of 95 °C for 15 s, 59 °C for

30 s, 72 °C for 33 s; and graded heating to 95 °C to generate dissociation curves to confirm amplification of a single PCR product. Primer pairs were designed using the NCBI/Primer-BLAST tool to identify primers unique for the intended target and were confirmed to amplify only a single product. Those used for amplification are available as Supplementary Table 1. Data were normalized to *Gapdh* as a housekeeping gene and analyzed by comparing C(t) values of control and G-CSF treated mice using the ΔΔC(t) method.

**DREADD experiments**. For these experiments, we utilized AAV2-hSyn-DIO-hM3Dq-mCherry (Gq-DREADD) purchased from the viral core at the University of North Carolina, Chapel Hill as well as the retrograding Canine adenovirus type 2 (CAV2)-Cre from the Institut de Génétique Moléculaire de Montpellier (Montpellier, France). Mice were anesthetized with ketamine (100 mg kg$^{-1}$) and xylazine (10 mg kg$^{-1}$) and positioned in a stereotactic frame. Ophthalmic ointment was placed in the eyes to prevent drying. A midline incision was made in the scalp and craniotomies were made using a dental drill. A 10 μl Nanofil Hamilton syringe (WPI, Sarasota, FL) with a 34-gauge beveled metal needle was used to infuse 0.5 μl virus at a rate of 100 nl/min. After the infusion, the needle was kept at the injection site for 10 min then slowly withdrawn. All animals received infusion of the CAV2-Cre virus in the NAc (From bregma: anteroposterior + 1.6 mm; mediolateral + 1.5 mm; dorsoventral −4.4 mm). Animals were then split into three experimental groups: (1) Control animals received no additional viral injections (2) mPFC-DREADD animals received an infusion of G$_q$-DREADD in the medial prefrontal cortex (From bregma: anteroposterior + 1.7 mm; mediolateral + 0.75 mm; dorsoventral −2.5 mm at 15°), and (3) VTA-DREADD animals received infusion of G$_q$-DREADD into the ventral tegmental area (From bregma: anteroposterior −3.3 mm; mediolateral +1.05 mm; dorsoventral −4.6 mm at an angle of 7°). Animals in all groups then received injections of clozapine-N-oxide (CNO) 1 mg kg$^{-1}$ i.p. daily for 7 days and animals were sacrificed for analysis. Expression of DREADD virus in the desired location was confirmed with qPCR for mCherry.

**Cocaine methiodide**. To assess the effects of peripheral cocaine on expression of G-CSF in both the periphery and the CNS we utilized the charge cocaine analog cocaine methiodide—which does not cross the blood brain barrier[25] to dissociate central vs. peripheral effects. For these experiments animals received daily injections of cocaine methiodide (20 mg kg$^{-1}$) or saline for 7 days and were then sacrificed 24 h after the final dose for analysis.

**Locomotor sensitization**. Locomotor sensitization was measured, as described[56]. Activity was monitored in rat-sized cages free of bedding that were set up in a locomotor apparatus designed to measure extent of ambulations in the $x$ and $y$ planes. On each of the assessment days the animals were given an injection of G-CSF (50 μg kg$^{-1}$, i.p.)[48] in their home cage 30 min prior to injection of saline or cocaine. On the first 2 days of each experiment the animals then received an i.p. injection of normal saline (10 ml kg$^{-1}$) and locomotor activity was monitored for 45 min. For each of the next 5 days animals were injected with 7.5 mg kg$^{-1}$ cocaine i.p. and their activity was monitored for 45 min.

**Conditioned place preference**. An unbiased conditioned place preference assay was carried out as described previously[56]. In brief, mice were evaluated for cocaine place preference using three chambered CPP Med Associates boxes and software. The two end chambers have distinct visual (gray vs. striped walls) and tactile (small grid vs. large grid flooring) cues to allow differentiation. On the pre-test day, animals were allowed to freely explore all three chambers for 20 min and those that showed a significant preference for one of the two chambers were excluded from further analysis (<10% of animals tested). Groups were then balanced and adjusted to balance out any pre-existing chamber bias. For G-CSF dose–response experiments, animals were injected each day with G-CSF (50 μg kg$^{-1}$, i.p.) in their home cage one hour prior to the start of any conditioning. For G-CSF neutralizing antibody experiments animals were injected with anti-G-CSF antibody (10 μg, i.p.) one hour before the start of each day, and minipump animals had anti-G-CSF infused into the NAc at a rate of 1 μg/side/day (details below). CPP was carried out by pairing an injection of saline with one chamber in the morning, and a second injection of cocaine (3.75, 7.5, 15 mg kg$^{-1}$) with the other chamber in the afternoon for two consecutive days. A cocaine dose of 7.5 mg kg$^{-1}$ was used for G-CSF antibody experiments. Conditioned place preference testing was carried out on the fourth day when each animal was again allowed to explore all chambers freely. Place preference score was taken as time on the cocaine paired side—time on the saline paired side.

**Minipump experiments**. Mice were surgically implanted with two subcutaneous Alzet minipumps (Durect corporation model 1007D) and bilateral guide cannulae (Plastics One) targeting the nucleus accumbens. The day before surgery cannulae were loaded with antibody solution and attached to pumps filled with either anti-G-CSF or pre-immune IgG. The pumps were set to each deliver 1 μg of antibody/day based on flow rate. Briefly, mice were anesthetized with Ketamine (100 mg kg$^{-1}$) and Xylazine (10 mg kg$^{-1}$) and an incision was made over the skull and the skin over the scapulae was blunt dissected from underlying fascia to allow for implantation of the pumps. Bilateral cannulae were delivered into the NAc

according to previously published coordinates (From bregma: anteroposterior, + 1.5; mediolateral, + 1.0; dorsoventral, −4.5)[57]. The cannulae were permanently fixed to the skull with Loctite adhesive, and cannulae tubing and pumps were all secured under the skin using several interrupted sutures.

**Sucrose preference**. To test for preference for natural rewards individually housed mice were allowed to habituate to two bottles of water for 1 day followed by three consecutive days of one bottle of water and one bottle of 1% sucrose. The bottles were weighed daily and the side was switched each day to prevent animals forming a preference for location. The ratio of sucrose/water intake was used as the measure of preference.

**Self-Administration surgery and training**. Rats and mice were anesthetized with ketamine (100 mg kg$^{-1}$) and xylazine (10 mg kg$^{-1}$) and implanted with chronic indwelling jugular catheters as described[31]. Animals were singly housed, and all sessions took place during the active/dark cycle (1200–1500 hours). After a 6-day recovery period, animals underwent training for self-administration where they were given access to a cocaine-paired lever on a fixed ratio one (FR1) schedule, which, upon responding, initiated an intravenous injection of cocaine (0.8 mg kg$^{-1}$ for rats, 0.5 mg kg$^{-1}$ for mice, infused over 5 s). After each response/infusion, the lever was retracted and a stimulus light was illuminated for a 20 s timeout period. A second lever (the inactive lever) was available; however, responding on this lever was recorded, but resulted in no programmed consequence. Acquisition was considered to have occurred when an animal allocated > 70% of their responses on the active lever (and > 20 injections) for two consecutive days and a stable pattern of infusion intervals was present. Subsequently animals went on to complete the threshold procedure (described in detail below) to assess how GCSF altered the motivational properties of cocaine.

**Threshold procedure**. Following acquisition of cocaine self-administration, the threshold procedure was used to determine differences in cocaine consumption and motivation between groups. A total of 20 animals underwent surgery and 4 were discarded to do catheter failure or failure to meet acquisition criteria. A total of 16 animals performed the threshold procedure. For this procedure, rats were injected with G-CSF (50 μg kg$^{-1}$) thirty minutes prior to each testing session[20,58]. The threshold procedure is a behavioral economics approach to assessing drug taking/seeking and reinforcing efficacy[28,29]. It consists of giving rats access to reinforcer while increasing the price, in effort, an animal must pay to obtain the reward. For the procedure with cocaine a descending series of 11 unit doses of cocaine (421, 237, 133, 75, 41, 24, 13, 7.5, 4.1, 2.4, and 1.3 μg/injection) are available on an FR1 schedule of reinforcement with no timeout. Each dose is available for 10 min, with each bin presented consecutively across the 110-min session. A similar approach was used for food reinforcement; however, increasing price was achieved by increasing the FR requirement (1, 2, 3, 5, 6, 8, 10, 14, 18, 25, 31, 44, 56, 78, 100) to obtain each food pellet. During this time, the lever is never retracted, and the only time out periods occur during the infusion of cocaine or the delivery of food. By plotting the consumption of cocaine or food as a function of price (number of responses need to obtain 1 mg of cocaine/food) demand curves are generated (Fig. 4) and behavioral economic principles can be applied to assess a variety of economic measures (described in detail below). During the initial bins of the procedure the dose is high, thus minimal effort is needed to obtain a preferred level of drug, i.e., the cost of cocaine is low. With decreasing doses the cost progressively increases and animals are required to exert more effort to obtain their preferred level of cocaine. Eventually, the dose becomes low enough that preferred levels of cocaine cannot be maintained and responding decreases. This point, where the cocaine level cannot be maintained (i.e., the first derivative point slope of the function = −1) is termed the P$_{max}$, or the maximal price paid. P$_{max}$ is directly correlated with traditional measures of reinforcing efficacy like progressive ratio responding[28,54]. Shifts in responding across the demand curve can be analyzed using behavioral economics principles, as described below. Responding during the first bin of the procedure is considered to reflect a loading phase and is not included in the analyses.

Behavioral economic analysis was used to determine the parameters of maximal price paid (P$_{max}$) and consumption at a minimally constraining price (Q$_0$), as described previously[28,59,60]. Briefly, P$_{max}$ and Q$_0$ values were derived mathematically using a demand curve. Demand curves were generated by curve-fitting individual animals' intake using an equation: $\log(Q) = \log(Q_0) + k \times (e - \alpha \times Q_0 \times C - 1)$[61,62]. In this equation, P$_{max}$ was determined to be the unit price at which the first derivative point slope of the function = $-1$[63]. The value k was set to two for all animals[61,62].

$Q_0$: $Q_0$ is a measure of the animals' preferred level of cocaine consumption. This can be measured when the dose is high and cocaine is available at low effort, or a minimally constraining price. This preferred level of consumption is established in the early bins of the threshold procedure.

P$_{max}$: Price is expressed as the responses emitted to obtain 1 mg of cocaine, thus as the dose is decreased in each consecutive bin of the threshold procedure, price increases. As the session progresses, animals must increase responding on the active lever in order to maintain stable intake. P$_{max}$ is the price at which the animal no longer emits enough responses to maintain intake and consumptions decreases.

Thus, animals with higher $P_{max}$ will increase responding to maintain cocaine levels farther into the demand curve; in other words, they will pay a higher price for cocaine. Previous work has demonstrated that $P_{max}$ is highly correlated with break points on a progressive ratio schedule of reinforcement, confirming that the threshold procedure accurately assesses reinforcing efficacy[29,60].

**Immunohistochemistry.** For these studies, we utilized two transgenic mouse lines. CX3CR1[GFP]/CCR2[RFP] mice express GFP in microglia and RFP in peripherally derived monocytes[64], and were generously provided by Dr. Miriam Merad. In addition, to identify cells expressing the D1-type dopamine receptor we utilized D1-tomato mice purchased from Jackson Labs (Stock #: 016204). Staining was performed using previously published methods[65]. Briefly, mice were transcardially perfused first with ice-cold PBS and then with a fixative solution containing 4% PFA paraformaldehyde (PFA). Brains were post-fixed for 24 h in 4% PFA at 4 °C. Sections of 30 µm thickness were cut in the frontal plane with a vibratome (Leica, Nussloch, Germany) and stored at −20 °C in a solution. Sections were permeabilized with 0.2% Triton X-100, followed by blocking with 35 normal donkey serum. Primary antibodies against G-CSF (Abcam, ab181053, 1:500 dilution), GCSF-Receptor (Life Science, LS-C393437, 1:100 dilution) and ds-Red/tdTomato (Santa Cruz Biotechnology, sc-390909, 1:500) were diluted in blocking solution and sections incubated overnight at 4 °C with gentle shaking. Sections were then incubated with secondary antibodies (donkey anti-mouse or anti-rabbit Alexa Fluor 488, donkey anti-goat Rhodamine Red, donkey anti-rabbit Cy5; Jackson ImmunoResearch, 1:500 dilution) for 2 h at room temperature. After washing, sections were incubated for 5 min with DAPI (NucBlue, Invitrogen) to achieve counterstaining of nuclei before mounting in Prolong Gold (Invitrogen). Protein expression was assessed in the nucleus accumbens using a LSM 710 laser-scanning confocal microscope (Carl Zeiss) imaged using a ×20, ×40 or ×63 objective with a 1.0 digital zoom.

**Statistical analysis.** All statistical analysis was performed using GraphPad Prism. Pairwise comparisons were performed using a two-tailed Student's $t$-test with Welch's correction when appropriate. One factor comparisons were performed using one-way ANOVA and Sidak's post-hoc tests. 2 × 2 comparisons were performed using two-way ANOVA with repeated measures and Holm-Sidak's post-hoc tests. Correlational analyses were performed using Pearson's correlation analysis.

**Data availability.** The data that support the findings of this study are available from the corresponding author upon reasonable request.

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

## Acknowledgements

This work was supported by NIH grants DA044308 to D.D.K., P01-DA008227 to E.J.N., and DA042111 to E.S.C., as well as by funds from the Brain and Behavior Research Foundation to E.S.C and D.D.K., and the Leon Levy Foundation and Seaver Family Foundation to D.D.K. We would like to thank the Human Immune Monitoring Core at Mount Sinai for providing their expertize in running the cytokine multiplex assays. We would like to thank Miriam Merad and Maria Casanova-Acebes for providing transgenic mice. We would also like to thank Erik Oleson for providing expertize and code to complete the behavioral economics experiments.

## Author contributions

E.S.C, A.G., Y.L.H., S.J.R., E.J.N. and D.D.K. designed experiments. E.S.C., A.G., E.G.P., M.S., N.L.M., J.A.L. and D.D.K. performed experiments and data analysis. E.S.C., A.G., E.G.P., M.S. and D.D.K. created the figures. E.S.C., A.G. and D.D.K. wrote the manuscript. All authors provided revisions and critical feedback on the final draft of the manuscript.

## Additional information

**Competing interests:** The authors declare no competing financial interests.

