## [Peer Review File · Nature Communications]

Reviewers' comments:

Reviewer #1 (Remarks to the Author):

In this study Calipari et al identify G-CSF as modulator of neuronal activity and behavior in response to cocaine. They identified G-CSF in a screen for changes in serum chemokines and cytokines following two modes of cocaine administration (ip and self-administration). They then show that treatment with G-CSF increases activity within reward circuits (using c-fos mRNA) and alters behavior (increased place preference, increased motivation for self-administration, enhanced lever pressing in the progressively increased cocaine price). They continue to show that both central and peripheral G-CSF manipulation can affect addictive behavior. Centrally, they demonstrate that blocking G-CSF in the NA (using neutralizing Ab abrogate some of the G-CSF effects on behavior and peripherally, they show that G-CSF injection affected the motivation for cocaine. This is an important study as they introduce immune mediator as potential therapeutics for addiction. It builds on an extensive literature connecting immune modulation with behavior. Nevertheless, there some aspects to this manuscript that needs further inquiry:

Major concerns:

1.The authors demonstrate that cocaine intake increase G-CSF serum levels, as well as mRNA expression levels in the NAc. However, it is not clear which cell types (in the brain and in the periphery) produce G-CSF.

2.It is also unclear which cells respond to G-CSF in the NAc, prefrontal cortex and in the periphery (is it a direct effect?)

3.The manuscript lacks a mechanistic understanding into the connections between cocaine administration and G-CSF levels. In other words, what are the cellular mechanisms connecting between exposure to cocaine and G-CSF expression levels? What is the signal? Is it cocaine itself?

4.G-CSF is known to increase cell mobilization, is there increased immune infiltration to the brain? This further highlights the need to identify the need to determine what is the source of G-CSF.

5.The selection of G-CSF for in depth analysis is fully justified in the study, nevertheless, it is unclear whether other cytokines will have similar effects? For example, David Engblom studies establish a link between prostaglandin, IL-1 and other pro-inflammatory mediators and alcohol intake and other potentially related behaviors.

6.The finding that locally blocking G-CSF in the NAc prevents association in the CPP paradigm is interesting. Can the authors show such abolishment of performance in another reward-associated behavioral paradigm? For example, the Sucrose Preference Test?

7.Pro-inflammatory mediators induce sickness behavior. Is it possible that some of the observed effects are related?

Minor concerns:

1.The paragraph regarding the threshold procedure (line 213- 238) is not clear. The terminology is confusing (for example: what is the difference between cost (line 217) and price (line 227)?).

2.The sentence in Lines 308-309 is also not clear.

3.Peripheral neutralization of G-CSF was not significant, but higher doses of the Ab were not tested.

Reviewer #2 (Remarks to the Author):

In this manuscript, Calipari and colleagues used rodent models to investigate the role of innate immune effectors in cocaine-related behaviors. They first used experimenter delivered and self-administered cocaine paradigms and subsequent serum analysis to identify increased granulocyte-colony stimulating factor (G-CSF) levels after cocaine exposure. They also found a significant correlation between G-CSF levels and both cocaine sensitization and daily cocaine intake. Through additional experiments, the authors showed that G-CSF administration enhanced cocaine-induced cfos in the nucleus accumbens and prefrontal cortex, and enhanced cocaine's locomotor sensitization and conditioned place preference effects. These enhancement effects were in contrast to the attenuation of cocaine place preference that was observed following nucleus accumbens infusion of a G-CSF neutralizing antibody. The authors also used a behavioral economic based threshold procedure, combined with cocaine self-administration, to demonstrate that G-CSF exposure enhances motivation for cocaine.

Overall, this is a well written manuscript on an interesting data set that has the potential to strongly impact the field. Importantly, the statistical approaches also appear to have been performed appropriately. The strengths of the work include the authors' use of a screening tool across 2 cocaine behaviors, and the confirmation of G-CSF's role through subsequent G-CSF manipulations in multiple behavioral assays. However, there are concerns regarding some aspects of the experimental design that are not addressed in the current version of the manuscript. The claims about the lack of G-CSF abuse potential also seem somewhat premature, given the lack of a more thorough examination of either G-CSF's potentially rewarding effects or its potential ability to also enhance motivation for natural rewards.

Specific concerns are described in detail below:

1. For the G-CSF and G-CSF receptor expression experiments in Figure 2i and 2j, the authors did not provide a rationale for why they did not use chronic cocaine exposure paradigms consistent with the sensitization and self-administration paradigms in Figure 1. Since the G-CSF effects were most pronounced after cocaine self-administration (Figure 1g), it is somewhat surprising that the authors did not examine G-CSF expression after cocaine self-administration. As appropriately referenced by the authors in the discussion, the prefrontal cortex contributes to the motivational and decision making components of cocaine-associated behavior. It would be helpful for the authors to acknowledge that the

lack of PFC effects may have been due to the use of 7 day experimenter delivered cocaine, and that the PFC effects may have emerged after self-administered cocaine exposure.

2. For Figure 3a, the authors should comment on the apparent lack of cocaine locomotor activating effects of cocaine injections 1 and 2. Initial cocaine activation has been previously observed by others, even after 2 days of saline injections. However, in Figure 3a it appears that the PBS group showed a delayed cocaine locomotor activation and sensitization phenotype.

3. For the place preference experiments, the authors include an important control to demonstrate that the G-CSF injection protocol alone does not alter place preference (Figure 3c). However, this experiment does not directly support the authors' claim that G-CSF is not rewarding, as the G-CSF injection is never associated with a specific chamber (as I understand the authors' protocol). To directly address the G-CSF reward question in the place preference paradigm, an experiment would need to be performed with G-CSF injections paired with 1 chamber during conditioning sessions.

4. In the discussion, the authors make a statement about G-CSF modulators having no abuse potential. However, this question is not extensively examined in this manuscript. From the authors' work, it is unclear whether the G-CSF enhanced motivation for reward is specific to cocaine or would also occur with natural rewards. It seems that the threshold procedure would be a useful paradigm to determine whether G-CSF non-specifically enhances motivation for rewards in general. Such an experiment could provide additional insight concerning G-CSF abuse potential as an agent that non-specifically enhances motivation for rewards.

5. Related to the authors' proposed utilization of G-CSF modulators as a pharmacological intervention, most of the experiments in the paper demonstrate enhanced cocaine-related behavior after G-CSF exposure. The authors' experiments did not demonstrate many effective manipulations to decrease cocaine-related behavior, other than the nucleus accumbens infusion experiment, as would likely be utilized as a pharmacotherapy. As such, the authors may want to reconsider their emphasis of the pharmacotherapeutic applications of their findings.

Minor comments:

6. In the results section and the figure legends, it would be helpful if the authors' clarified which experiments were performed in mice and which experiments were performed in rats.

7. The Figure Legend for Figure 1 contains a typographical error. Panel label "G" is included twice, and panel label "J" has not been included.

Reviewer #3 (Remarks to the Author):

NCOMMS-17-09548

Granulocyte-colony stimulating factor controls neural and behavioral plasticity in response to cocaine

The present manuscript by Calipari, Godino and colleagues identifies G-CSF as a potent mediator of cocaine-induced adaptations. The authors show that treatment with G-CSF increased cocaine-induced activation of reward circuits, increased cocaine place preference, enhanced motivation to self-administer cocaine, and enhanced lever pressing for low probability cocaine. Infusion of G-CSF neutralizing antibody into the nucleus accumbens prevented these effects, which provides a direct link between central actions of G-CSF and cocaine reward. In addition, they show that systemic injections of G-CSF were sufficient to alter the motivation for cocaine without abuse potential.

Although these results are interesting and the claims are appropriately discussed in the context of previous literature, there are important issues related to the Methods and Results sections that diminish its overall impact. The authors are encouraged to attend to the following concerns:

1) Given the exploratory nature of the data obtained from multiplex analysis with several analytes (Tables S1 and S2), it is highly recommended to conduct a test for multiple comparisons. As the number of comparisons increases, it becomes more likely that the groups being compared (cocaine vs. control) will appear to differ in terms of at least one analyte (cytokines, chemokines or growth factors) due to random sampling error alone. There is no universally accepted approach for dealing with the problem of multiple comparisons but the classic approach is to control the familywise error rate (e.g., Bonferroni correction). An alternative approach is to control the false discovery rate (e.g., Benjamini-Hochberg procedure).

2) The authors performed a 2-way ANOVA (using saline/cocaine and PBS/G-CSF as factors) for c-Fos expression in the PFC, NAc, BLA, CPu and VHip (Fig. 2b-f). The primary purpose of this statistical procedure is to understand whether there is an interaction between factors and, consequently to describe this interaction effect using post hoc tests for multiple comparisons or simple effects tests. The post hoc tests have been revised for clarity in the text and figure.

Related to this suggestion, Which is the reason why authors represent G-CSF vs PBS (% change cFos) of the cocaine group? This is only my supposition because the cocaine group is not indicated in graphs of Fig. 2b-f.

3) The authors tested a range of cocaine doses (3.75, 7.50 and 15 mg/kg, ip) in mice treated with G-CSF or PBS using a conditioned place preference paradigm. Please, justify the reason why a 2-way ANOVA was not used to compare preference data among doses in both G-CSF and PBS subgroups (Fig. 3b).

4) In the Fig. 4d, it is not clear what (*) and (***) symbols denote because the figure legend uses the same symbols for main effects (and/or interaction) and post hoc tests. I have to suppose that both symbols are referred to the post hoc differences between PBS and G-CSF. Please, describe the post hoc tests in the text.

5) The averaged demand curves from both groups (PBS and G-CSF) are shown in Fig. 4g and the statistical analysis is well described in the figure legend (lines 612-614). However, although there were main effects of price and treatment, only the main effect of treatment was reported in the text. Were post hoc tests performed for these data?

6) All experiments were conducted in mice with the exception of the threshold procedure, which was performed in rats self-administering cocaine. Why was the threshold procedure

not performed in mice?

7) Although the dose of G-CSF for locomotor sensitization in mice was chosen based on Tsai et al., 2007 (50 µg/kg, ip), there is no information for rats. Please, justify this dose for rats.

Minor comments

- The names of statistical tests (including post hoc tests) and their description should be indicated in Methods section (Statistical analysis). Please, remove these names from the text of the Results section (e.g., lines 85, 128...). Furthermore, you can find them in the Figure legends.

- The authors have to be constant with the description of statistics, df (degrees of freedom) and p-value throughout the text (e.g., line 236 vs. line 241). Furthermore, the use of df is inconsistently reported in the Results section (text and figure legends).

- Please, explain "...antibiotic treated mice..." in line 413.

- The authors have to indicate that Gapdh was used as housekeeping gene in the study. This has to be unequivocally indicated in the manuscript. How were primer pairs chosen for amplification?

- Line 521 establishes: "...p values are reported as * <0.05 , ** <0.01 & *** <0.001 ." However, in Fig. 2c you can see ****. Please, define it.

- Threshold procedure is extensively described in comparison with other chapters in the Methods section (lines 469-515) and it includes several references. Consider reducing this chapter.

- The dose of G-CSF for locomotor sensitization experiments (50 µg/kg) is indicated in the Results section (line 419) but not in the Methods section or figure legend (Fig. 4). This information has to be added in the Methods.

We would like to thank all three reviewers for their detailed and insightful comments on our manuscript. To address these comments and concerns we have performed extensive additional experiments, and we feel that the manuscript is substantially stronger with the additional data and discussion.

To this revision, we have added the following new experimental findings:

- We define activation of the mPFC → NAc, but not VTA → NAc, pathway as the central mechanisms by which G-CSF levels are increased within the nucleus accumbens (NAc).
- Using cocaine-methiodide we rule out peripheral actions of cocaine as a factor in contributing to the increases in central G-CSF levels.
- Using food-maintained operant responding we show that the effects of G-CSF on the motivational properties of cocaine are specific for drug rewards.
- Using sucrose preference, we show that G-CSF does not alter natural reward consumption.
- We define the localization of G-CSF and its receptor in the NAc using immunohistochemistry.
- Finally, we provide further validation that G-CSF does not have effects on motivation on its own and acts to modulate the motivational properties of drug rewards.

In addition to these experimental additions, we have made extensive revisions to the text of the manuscript to improve clarity and accuracy (significant text changes are presented in **blue font** throughout the manuscript). We remain fully confident that it makes a very significant contribution to the field and hope that you will agree. We thank you again for your time and consideration. Point-by-point responses to all reviewer comments are below.

Reviewer #1:

Major concerns:

1. The authors demonstrate that cocaine intake increase G-CSF serum levels, as well as mRNA expression levels in the NAc. However, it is not clear which cell types (in the brain and in the periphery) produce G-CSF.

This is an excellent question. The sources of G-CSF in the periphery are myriad and well characterized, and include monocytes, macrophages, endothelial cells and fibroblasts (e.g. PMIDS: 2787682, 3257150, 2439155, 10081506, 1720034). Findings in brain also suggest diverse expression of G-CSF including in neurons (e.g. PMIDS: 16007267, 16839644), microglia (e.g. PMIDS: 28646409, 16049425), as well as astrocytes (e.g. PMIDS: 28526328, 25448005). A study in human brain found G-CSF and its receptor to be expressed at varying densities in both neurons and glia throughout the brain from cortex through brainstem (PMID: 28732710).

To address the reviewer's question experimentally, we performed immunostaining for both G-CSF and its receptor and obtained results consistent with a pattern of expression in multiple cell types (**Figure S1**). As can be seen in this figure, G-CSFR is expressed in both D1⁺ and D2⁺ medium spiny neurons, but also in non-neuronal cells as well. Staining for G-CSF itself demonstrated a similar pattern of staining, but with less clear labeling of the MSN populations. Based on these data it is clear that the expression pattern of G-CSF and its receptor in the brain is similarly heterogeneous to the pattern seen in the periphery. We have added additional discussion within the manuscript about this important point.

2. It is also unclear which cells respond to G-CSF in the NAc, prefrontal cortex and in the periphery (is it a direct effect?)

This is another important question. The current manuscript identifies G-CSF as: (1) a cytokine regulated by cocaine in both serum and brain in multiple paradigms, (2) a modulator of neuronal plasticity in response to cocaine with identification of specific neural pathways in which it is activated, and (3) a potent regulator of behavioral responses to cocaine in multiple behavioral paradigms including self-administration, but now with clarification that this is a cocaine-specific phenomenon. The next step in future manuscripts will be to define the complex interplay between neurons and the cellular microenvironment that lead to these effects, but such work is beyond the scope of the present manuscript.

As above, it seems that the cells that express the G-CSF receptor are diverse and include the D1⁺ and D2⁺ populations of NAc medium spiny neurons as well as glial cells. Thus, the determination of how G-CSF signaling in different cell types affects plasticity in response to cocaine will be a complicated project involving viral manipulations of multiple targets in multiple cell types, and likely generation of new transgenic mouse

lines to allow for a full and rigorous answer to this question. This work will be the subject of our next series of experiments over the course of the next several years.

3. The manuscript lacks a mechanistic understanding into the connections between cocaine administration and G-CSF levels. In other words, what are the cellular mechanisms connecting between exposure to cocaine and G-CSF expression levels? What is the signal? Is it cocaine itself?

This comment is particularly important and prompted additional studies to define the precise mechanisms by which cocaine leads to increases in NAc G-CSF levels. To this end, we designed a two-part experiment to determine if central or peripheral actions of cocaine were critical in G-CSF increases. First, we increased peripheral cocaine levels using a cocaine analog (cocaine methiodide) that does not cross the blood-brain barrier. This was to determine if peripheral actions of cocaine, independent of its central actions, induced changes in cytokines peripherally that ultimately lead to increases in the NAc. The second experiment determined if increases in the activity of projection pathways, that were shown to be activated by cocaine in Figure 2, can increase central G-CSF levels independent of cocaine actions peripherally. Together these experiments help to pinpoint the locus of cocaine actions on G-CSF.

The first experimental group received daily injections of cocaine methiodide, a charged analogue of cocaine that does not cross the blood-brain barrier, as stated earlier. This allowed us to assess the effect of prolonged cocaine exposure on G-CSF expression in both the serum and in the brain in a system in which cocaine is segregated from the CNS. We find that cocaine-methiodide had no effect on central expression of G-CSF or on peripheral G-CSF serum levels. Previous experiments within the manuscript have shown the blocking peripheral G-CSF is not sufficient to block CPP, while blocking central G-CSF inhibited the formation of place preference. Thus, together, these experiments define central cocaine action as the critical factor contributing to increases in NAc G-CSF and its actions on reward and motivation.

To determine a potential mechanism by which cocaine's effects on the CNS act to drive increases in G-CSF, we also performed a series of experiments in which we used G_q-coupled DREADD expression to stimulate the mPFC→NAc or VTA→NAc pathways (by utilizing retrograde Cre in NAc and Cre-dependent DREADD in mPFC/VTA) in isolation. These pathways were chosen based on data from Figure 2 showing increases in c-FOS expression in these pathways in response to cocaine. Interestingly, stimulation of the PFC→NAc pathway only induces NAc induction of G-CSF and its receptor, with no significant effect on the periphery. In contrast, stimulating the VTA→NAc pathway had no effect, suggesting that the effects were pathway specific. Together, these data suggest that induction of NAc G-CSF is dependent on stimulation of very specific neural circuits induced by cocaine exposure.

4. G-CSF is known to increase cell mobilization, is there increased immune infiltration to the brain? This further highlights the need to identify the need to determine what is the source of G-CSF.

This is an important consideration given that the role of infiltrating immune cells has been implicated in models of depression and other psychiatric conditions. To answer this, we performed immunostaining in the NAc of Cx3cr1^{GFP}/Ccr2^{RFP} dual transgenic animals. These animals allow for labeling of microglia in green and peripherally derived monocytes in red. Animals were treated either with daily injections of saline or one week of daily G-CSF (50 µg/kg) injections, then perfused and imaged. As shown in **Figure S2**, there is no significant infiltration of RFP-positive peripheral cells in either treatment condition.

5. The selection of G-CSF for in depth analysis is fully justified in the study, nevertheless, it is unclear whether other cytokines will have similar effects? For example, David Engblom studies establish a link between prostaglandin, IL-1 and other pro-inflammatory mediators and alcohol intake and other potentially related behaviors.

This is an important point, and we certainly did not mean to imply that other cytokines would not affect behavior in a similar manner. Indeed, work from the Stellwagen group and the Watkins group have shown roles for TNF-α and IL-1β, respectively, in behavioral response to cocaine. We have added additional text to the discussion to highlight these studies and those from the Engblom group to highlight the need for further characterization of the role of many other cytokines in behavioral response to drugs of abuse.

6. The finding that locally blocking G-CSF in the NAc prevents association in the CPP paradigm is interesting. Can the authors show such abolishment of performance in another reward-associated behavioral paradigm? For example, the Sucrose Preference Test?

We consider this to be a very important issue and have performed several experiments to address the role of G-CSF in affecting responses to natural rewards. We first performed a two-bottle sucrose preference test to measure how injections with G-CSF would affect sucrose choice and consumption. Interestingly, daily injections with G-CSF did not affect sucrose preference in any meaningful way (**Figure S3a**). Additionally, to get a more nuanced understanding of how increased levels of G-CSF affected responses to natural reward, we also performed a behavioral economics threshold task for food in G-CSF-treated and control rats – similar to what was done for cocaine. Here, we also found that treatment with G-CSF did not significantly affect responding for food reward. These data suggest that the behavioral effects of G-CSF are more specific to modulating behavioral response to drugs of abuse than to natural rewards.

7. Pro-inflammatory mediators induce sickness behavior. Is it possible that some of the observed effects are related?

This was something we considered early on, but think it is unlikely for a number of reasons. Animals injected with this concentration of G-CSF show no changes in bodyweight even with prolonged injection (implying normal nutrition and hydration status). When animals are injected with repeated doses of G-CSF they do not show decreased locomotor activity. Likewise, injections of G-CSF do not lead to any detectable anhedonic-like behaviors and show normal intake of natural (food) and drug (cocaine) rewards. Importantly, while G-CSF is a cytokine that is capable of increasing mobilization of neutrophils from the bone marrow, it is often considered to be anti-inflammatory in that it decreases production of pro-inflammatory cytokines such TNF- α and IL-1 β in numerous previous studies.

Minor concerns:

1. The paragraph regarding the threshold procedure (line 213- 238) is not clear. The terminology is confusing (for example: what is the difference between cost (line 217) and price (line 227)?).

We understand that language used to describe behavioral economics is complicated, and to this end we have revised this section to clarify the procedure and have made sure to clearly define the terminology that is used throughout the manuscript.

2. The sentence in Lines 308-309 is also not clear.

This sentence has been clarified. Thank you.

3. Peripheral neutralization of G-CSF was not significant, but higher doses of the Ab were not tested.

Multiple doses of antibody were initially tested (5 μ g & 10 μ g), but behavioral response was not different between the two so the results of the higher dose were presented.

Reviewer #2:

... The claims about the lack of G-CSF abuse potential also seem somewhat premature, given the lack of a more thorough examination of either G-CSF's potentially rewarding effects or its potential ability to also enhance motivation for natural rewards.

An excellent point, and similar in nature to one raised by Reviewer #1. We have now added studies with G-CSF treatment during sucrose preference testing in mice, and in a behavioral economics threshold task for food rewards in rats. G-CSF did not produce significant effect in either paradigm, demonstrating that the effects of G-CSF are specific to drug rewards.

Specific concerns are described in detail below:

1. For the G-CSF and G-CSF receptor expression experiments in Figure 2i and 2j, the authors did not provide a rationale for why they did not use chronic cocaine exposure paradigms consistent with the sensitization and self-administration paradigms in Figure 1. Since the G-CSF effects were most pronounced after cocaine self-administration (Figure 1g), it is somewhat surprising that the authors did not examine G-CSF expression after cocaine self-administration. As appropriately referenced by the authors in the discussion, the prefrontal cortex contributes to the motivational and decision making components of cocaine-associated behavior. It would be helpful for the authors to acknowledge that the lack of PFC effects may have been due to the use of 7 day experimenter delivered cocaine, and that the PFC effects may have emerged after self-administered cocaine exposure.

The point about the difference in protocols is well taken. For Figure 1, the experimenter-administered cocaine was carried out for 10 days to directly match the length of exposure in the self-administration experiments as the levels were being directly compared to one another. The experiments in Figure 2 were carried out for 7 days – which is the more typical prolonged cocaine exposure paradigm used in the laboratory. We have added clarification of the difference in the regimens to the Methods, and have added a point about the lack of PFC effects at 7 days to the Discussion section.

2. For Figure 3a, the authors should comment on the apparent lack of cocaine locomotor activating effects of cocaine injections 1 and 2. Initial cocaine activation has been previously observed by others, even after 2 days of saline injections. However, in Figure 3a it appears that the PBS group showed a delayed cocaine locomotor activation and sensitization phenotype.

An excellent point. We have added clarification on this to both the Results and Discussion sections.

3. For the place preference experiments, the authors include an important control to demonstrate that the G-CSF injection protocol alone does not alter place preference (Figure 3c). However, this experiment does not directly support the authors' claim that G-CSF is not rewarding, as the G-CSF injection is never associated with a specific chamber (as I understand the authors' protocol). To directly address the G-CSF reward question in the place preference paradigm, an experiment would need to be performed with G-CSF injections paired with 1 chamber during conditioning sessions.

We agree that this is an important control, and we have added a CPP experiment in which G-CSF is paired with the P.M. chamber in the same manner as cocaine is in the other experiments. There was no significant preference or aversion formed to the G-CSF-paired chamber. These data are now included as panel **Figure 4d**.

4. In the discussion, the authors make a statement about G-CSF modulators having no abuse potential. However, this question is not extensively examined in this manuscript. From the authors' work, it is unclear whether the G-CSF enhanced motivation for reward is specific to cocaine or would also occur with natural rewards. It seems that the threshold procedure would be a useful paradigm to determine whether G-CSF non-specifically enhances motivation for rewards in general. Such an experiment could provide additional insight concerning G-CSF abuse potential as an agent that non-specifically enhances motivation for rewards.

An excellent point. We have now included a threshold experiment for food reward in rats treated with PBS or G-CSF. These data are now included as **Figure S3b,c** and demonstrate that treatment with G-CSF does not alter responding for food reward.

5. Related to the authors' proposed utilization of G-CSF modulators as a pharmacological intervention, most of the experiments in the paper demonstrate enhanced cocaine-related behavior after G-CSF exposure. The authors' experiments did not demonstrate many effective manipulations to decrease cocaine-related behavior, other than the nucleus accumbens infusion experiment, as would likely be utilized as a pharmacotherapy. As such, the authors may want to reconsider their emphasis of the pharmacotherapeutic applications of their findings.

This is a good point. To translate these findings to clinical interventions it will be particularly important to find pharmacological agents that inhibit G-CSF action and ultimately reduce the motivation for cocaine, which will be the goal of future studies within our lab and in collaboration with clinicians here at Mount Sinai. We have removed the strong wording and now just suggest that targeting G-CSF may provide a potential treatment strategy.

Minor comments:

6. In the results section and the figure legends, it would be helpful if the authors' clarified which experiments were performed in mice and which experiments were performed in rats.

This has been clarified.

7. The Figure Legend for Figure 1 contains a typographical error. Panel label "G" is included twice, and panel label "J" has not been included.

This has been corrected.

Reviewer #3 (Remarks to the Author):

1) Given the exploratory nature of the data obtained from multiplex analysis with several analytes (Tables S1 and S2), it is highly recommended to conduct a test for multiple comparisons. As the number of comparisons increases, it becomes more likely that the groups being compared (cocaine vs. control) will appear to differ in terms of at least one analyte (cytokines, chemokines or growth factors) due to random sampling error alone. There is no universally accepted approach for dealing with the problem of multiple comparisons but the classic approach is to control the familywise error rate (e.g., Bonferroni correction). An alternative approach is to control the false discovery rate (e.g., Benjamini-Hochberg procedure).

We completely agree that data obtained from multiplex analysis is exploratory and care should be made when making conclusions about data obtained from this method. We also agree that there is no universally accepted approach for dealing with the problem of false discovery rate, thus, our approach was to further validate the significantly-regulated cytokines in a number of ways. The first was to correlate their expression levels with cocaine-mediated behaviors. This ruled out things that may be spuriously increased and not related to cocaine exposure. Second, we only followed up on the cytokine that was changed in the same direction in two independent experiments. Third, in these two experiments the cytokine had to be correlated with cocaine-associated behaviors. Finally, we performed a number of experiments to causally link G-CSF to cocaine-associated behaviors (Figures 2-5). In each of these independent experiments we showed that G-CSF was increased, correlated with cocaine sensitization, correlated with voluntary cocaine consumption, could act to increase the motivation for drug rewards, and that decreasing central G-CSF levels was capable of blocking CPP. Thus, we feel that while multiplex analysis does have some statistical limitations in detecting significant factors, we have ruled out this as a false positive through many additional criteria and experiments.

2) The authors performed a 2-way ANOVA (using saline/cocaine and PBS/G-CSF as factors) for c-Fos expression in the PFC, NAc, BLA, CPu and VHip (Fig. 2b-f). The primary purpose of this statistical procedure is to understand whether there is an interaction between factors and, consequently to describe this interaction effect using post hoc tests for multiple comparisons or simple effects tests. The post hoc tests have been revised for clarity in the text and figure. Related to this suggestion, Which is the reason why authors represent G-CSF vs PBS (% change cFos) of the cocaine group? This is only my supposition because the cocaine group is not indicated in graphs of Fig. 2b-f.

We have clarified our reporting of the two-way ANOVA and post-hoc statistics. The additional panels in this figure were meant to add clarity - to show that there were differences in cocaine actions in the presence or absence of G-CSF. Since it seems they may be having the opposite effect we have now removed these panels from the manuscript.

3) The authors tested a range of cocaine doses (3.75, 7.50 and 15 mg/kg, ip) in mice treated with G-CSF or PBS using a conditioned place preference paradigm. Please, justify the reason why a 2-way ANOVA was not used to compare preference data among doses in both G-CSF and PBS subgroups (Fig. 3b).

These statistics were updated in the results section and figure legends.

4) In the Fig. 4d, it is not clear what () and (***) symbols denote because the figure legend uses the same symbols for main effects (and/or interaction) and post hoc tests. I have to suppose that both symbols are referred to the post hoc differences between PBS and G-CSF. Please, describe the post hoc tests in the text.*

This has been clarified in each figure.

5) The averaged demand curves from both groups (PBS and G-CSF) are shown in Fig. 4g and the statistical analysis is well described in the figure legend (lines 612-614). However, although there were main effects of price and treatment, only the main effect of treatment was reported in the text. Were post hoc tests performed for these data?

The text has been updated to include reporting of both main effects. The description of the post-hoc tests has been clarified in the figure legend.

6) All experiments were conducted in mice with the exception of the threshold procedure, which was performed in rats self-administering cocaine. Why was the threshold procedure not performed in mice?

Self-administration is notoriously difficult to do in mice even for simple tasks. Complicated tasks such as the threshold procedure are not well established in mice. Thus, to measure motivation in a volitional task, rats must be used.

7) Although the dose of G-CSF for locomotor sensitization in mice was chosen based on Tsai et al., 2007 (50 µg/kg, ip), there is no information for rats. Please, justify this dose for rats.

We apologize for the oversight. Similar weight-based dosing has been used in a number of studies in rats as well (e.g. PMIDs: 25585014, 25816736, 27680311). We have added this detail to the Methods section.

Minor comments

- The names of statistical tests (including post hoc tests) and their description should be indicated in Methods section (Statistical analysis). Please, remove these names from the text of the Results section (e.g., lines 85, 128...). Furthermore, you can find them in the Figure legends.

This has been corrected.

- The authors have to be constant with the description of statistics, df (degrees of freedom) and p-value throughout the text (e.g., line 236 vs. line 241). Furthermore, the use of df is inconsistently reported in the Results section (text and figure legends).

This has been corrected.

- Please, explain "...antibiotic treated mice..." in line 413.

This was a mistake and this has been removed.

- The authors have to indicate that Gapdh was used as housekeeping gene in the study. This has to be unequivocally indicated in the manuscript. How were primer pairs chosen for amplification?

This has been corrected. A description of primer pair selection has been added to the Methods.

*- Line 521 establishes: "...p values are reported as *<0.05, **<0.01 & ***<0.001." However, in Fig. 2c you can see ****. Please, define it.*

This has been defined.

- Threshold procedure is extensively described in comparison with other chapters in the Methods section (lines 469-515) and it includes several references. Consider reducing this chapter.

This has been revised for clarity.

- The dose of G-CSF for locomotor sensitization experiments (50 µg/kg) is indicated in the Results section (line 419) but not in the Methods section or figure legend (Fig. 4). This information has to be added in the Methods.

This has been corrected.

Reviewers' comments:

Reviewer #1 (Remarks to the Author):

We thank the authors for carefully addressing all the comments. The manuscript is clearly improved. There are few other issues that require some clarification.

For comments 1 and 2 that were related questions and I am not sure that they were fully addressed because there was no comparison between the cocaine-injected and control to identify the cells that can account the difference observed in G-CSF. Nevertheless, the addition provided by the authors is also helpful.

For comment 4. "G-CSF is known to increase cell mobilization, is there increased immune infiltration to the brain? This further highlights the need to identify the need to determine what is the source of G-CSF. " The response to the question is very helpful. Although it is interesting (not necessary) to see how cocaine itself affected the infiltration.

For comment 6. "The finding that locally blocking G-CSF in the NAc prevents association in the CPP paradigm is interesting. Can the authors show such abolishment of performance in another reward-associated behavioral paradigm? For example, the Sucrose Preference Test? " The added experiments are very important and given that they are somewhat surprising, require a more extensive discussion in comparison to other mechanisms that have similar distinctive effects.

Reviewer #2 (Remarks to the Author):

The authors have sufficiently addressed my concerns. The revised manuscript, with the included new experiments, is much improved.

Reviewer #3 (Remarks to the Author):

The authors have addressed all of my comments. I recommend to accept the paper.

We are excited to see that the reviewers unanimously agreed that the additional experiments and revisions greatly improved the manuscript. There were a small number of additional minor comments that we have addressed specifically below as well as within the manuscript. Further, we have again added additional experiments to address these concerns and we now hope that the manuscript is suitable for publication.

As previously, significantly changed text within the manuscript document has been colored blue for ease of reading.

Reviewer #1:

For comments 1 and 2 that were related questions and I am not sure that they were fully addressed because there was no comparison between the cocaine-injected and control to identify the cells that can account the difference observed in G-CSF. Nevertheless, the addition provided by the authors is also helpful. This is an important point and we have now conducted an additional experiment to address this specific question. This is included within the manuscript as revised **Supplemental Figure S1**. Briefly, mice were treated with cocaine for 7 days and then immunohistochemistry was done to determine if there were apparent cell-type specific changes in the distribution of G-CSF and its receptor. While we found changes in the relative levels of G-GCSF and its receptor in other experiments within the manuscript there were not any apparent changes in the distribution between cell types in the NAc following cocaine treatment.

For comment 4. "G-CSF is known to increase cell mobilization, is there increased immune infiltration to the brain? This further highlights the need to identify the need to determine what is the source of G-CSF. " The response to the question is very helpful. Although it is interesting (not necessary) to see how cocaine itself affected the infiltration. We agree that this was an important and interesting point and were glad to have addressed the effects of G-CSF in the previous revision. With regard to the cocaine effects, this is also interesting and we plan to do future studies to look at the effects of cocaine on immune cell infiltration within the brain.

For comment 6. The added experiments are very important and given that they are somewhat surprising, require a more extensive discussion in comparison to other mechanisms that have similar distinctive effects. We agree that these findings are very surprising and that additional discussion is needed. We have now added a paragraph to the discussion outlining the differences between food and drug reinforcement and why there could be differential effects in that context.

REVIEWERS' COMMENTS:

Reviewer #1 (Remarks to the Author):

Thank you for addressing most of the issues that were raised in the review.